# Stepwise evolution of *Salmonella* Typhimurium ST313 causing bloodstream infection in Africa

Caisey V. Pulford [1], Blanca M. Perez-Sepulveda[1], Rocío Canals [1], Jessica A. Bevington [1], Rebecca J. Bengtsson[1], Nicolas Wenner[1], Ella V. Rodwell[1], Benjamin Kumwenda [2], Xiaojun Zhu [1], Rebecca J. Bennett [1], George E. Stenhouse [1], P. Malaka De Silva[1], Hermione J. Webster[1], Jose A. Bengoechea [3], Amy Dumigan [3], Alicia Tran-Dien[4], Reenesh Prakash[5], Happy C. Banda[5], Lovemore Alufandika[5], Mike P. Mautanga[5], Arthur Bowers-Barnard[1], Alexandra Y. Beliavskaia[1], Alexander V. Predeus [1], Will P. M. Rowe[1], Alistair C. Darby[1], Neil Hall[6], François-Xavier Weill [4], Melita A. Gordon[5], Nicholas A. Feasey[5], Kate S. Baker [1] and Jay C. D. Hinton [1]✉

Bloodstream infections caused by nontyphoidal *Salmonella* are a major public health concern in Africa, causing ~49,600 deaths every year. The most common *Salmonella enterica* pathovariant associated with invasive nontyphoidal *Salmonella* disease is *Salmonella* Typhimurium sequence type (ST)313. It has been proposed that antimicrobial resistance and genome degradation has contributed to the success of ST313 lineages in Africa, but the evolutionary trajectory of such changes was unclear. Here, to define the evolutionary dynamics of ST313, we sub-sampled from two comprehensive collections of *Salmonella* isolates from African patients with bloodstream infections, spanning 1966 to 2018. The resulting 680 genome sequences led to the discovery of a pan-susceptible ST313 lineage (ST313 L3), which emerged in Malawi in 2016 and is closely related to ST313 variants that cause gastrointestinal disease in the United Kingdom and Brazil. Genomic analysis revealed degradation events in important virulence genes in ST313 L3, which had not occurred in other ST313 lineages. Despite arising only recently in the clinic, ST313 L3 is a phylogenetic intermediate between ST313 L1 and L2, with a characteristic accessory genome. Our in-depth genotypic and phenotypic characterization identifies the crucial loss-of-function genetic events that occurred during the stepwise evolution of invasive *S*. Typhimurium across Africa.

O ver the past decade, bloodstream infections (BSI) caused by nontyphoidal *Salmonella* (NTS) have killed approximately 650,000 people globally[1]. Invasive NTS (iNTS) disease disproportionately affects immunocompromised individuals such as adults with human immunodeficiency virus and children under five years of age with malaria, malnutrition or severe anaemia living in lower-to-middle-income countries[2]. Recently, a systematic review established that *Salmonella* was the most frequently isolated pathogen in hospitalized patients diagnosed with community-onset BSI in Africa and Asia (2008 to 2018)[3]. The *S. enterica* serovar Typhimurium (*S.* Typhimurium) is responsible for approximately two-thirds of iNTS disease cases in Africa[4], and consequently is a focal point for research.

The majority of *S.* Typhimurium isolated from BSI in Africa have sequence type ST313[5]. There are two distinct and tightly clustered lineages of ST313, which differ from each other by 455 single nucleotide polymorphisms (SNPs)[5]. ST313 lineage 1 (L1) and ST313 lineage 2 (L2) differ by 700 SNPs[6] from ST19, which commonly causes gastrointestinal infections globally. The African ST313 L1 and L2 are genetically distinct from ST313 found in the United Kingdom and Latin America[7,8].

It has been proposed that antimicrobial resistance (AMR) contributed to the success of ST313 lineages in Africa, with a large proportion of strains being multidrug resistant (MDR)[9]. Amongst *S.* Typhimurium ST313, chloramphenicol resistance is specifically found in L2, and is thought to have contributed to the clonal replacement of L1 by L2 prior to 2001, when chloramphenicol was the empirical treatment choice for suspected sepsis in Malawi[6]. Subsequently, chloramphenicol was replaced by the oral fluoroquinolone ciprofloxacin for treatment of iNTS infections in several parts of Africa including Malawi. The third-generation cephalosporin ceftriaxone became the empirical therapeutic for suspected sepsis from 2005[9]. More recently, variants of ST313 L2 with resistance to third-generation cephalosporins and ciprofloxacin or azithromycin have been identified in Malawi and the Democratic Republic of Congo, respectively[10,11].

In addition to AMR, the ST313 lineages have accumulated loss-of-function mutations (genome degradation)[12]. Evidence of host adaptation in ST313 involved specific pseudogenes that inactivated gene function and modified metabolic and virulence phenotypes[5,13–18]. The functional relevance of genome degradation in ST313 has been investigated at the transcriptomic, proteomic and phenotypic levels for several key isolates[5,13–17,19]. To understand the evolutionary trajectory and epidemiological relevance of functional degradative events, a study of *Salmonella* genomes from a wide range of contemporary and historical isolates is required.

[1]Institute of Infection, Veterinary and Ecological Sciences, University of Liverpool, Liverpool, UK. [2]University of Malawi, College of Medicine, Blantyre, Malawi. [3]Wellcome–Wolfson Institute for Experimental Medicine, Queen's University Belfast, Belfast, UK. [4]Institut Pasteur, Unité des Bactéries Pathogènes Entériques, Paris, France. [5]Malawi–Liverpool–Wellcome Trust Clinical Research Programme, Blantyre, Malawi. [6]Earlham Institute, Norwich Research Park, Norwich, UK. ✉e-mail: jay.hinton@liverpool.ac.uk

Here we study a large and up-to-date collection of ST313, revealing the stepwise evolution of *S.* Typhimurium causing BSI in Africa. We uncover a pan-susceptible lineage (ST313 L3) with a distinct genome-degradation pattern, which emerged in Malawi in 2016. Despite the recent appearance of ST313 L3 in clinics, our core and accessory genome analyses reveal it to be a phylogenetic intermediate between ST313 L1 and L2. We perform genotypic and phenotypic characterization of the key loss-of-function events that occurred along the evolutionary pathway of invasive *S.* Typhimurium across Africa.

## Results

**Assembling an informative collection of *S.* Typhimurium isolates.** A combination of historical and contemporary *S.* Typhimurium bloodstream isolates was sampled from two sources (Fig. 1). The main dataset consisted of 608 human bloodstream isolates collected between 1996 and 2018 from the Malawi–Liverpool–Wellcome Trust Clinical Research Programme (MLW) in Blantyre, Malawi. To provide context, 72 human bloodstream isolates collected between 1966 and 2012 by the Unité des Bactéries Pathogènes Entériques, Institut Pasteur, Paris, France were also included. This contextual collection was derived from 13 countries in Africa, 1 in Europe and 1 in Asia. A description of the isolates is summarized in Fig. 1, details of sequencing quality control are available in Supplementary Table 1 and the complete metadata, including genome accession numbers, are in Supplementary Table 2.

**Population structure of *S.* Typhimurium.** To define the population structure of *S.* Typhimurium currently causing BSI in Africa, we explored the relationship of the 680 *S.* Typhimurium genomes from the MLW and contextual collections. The two predominant sequence types causing BSI in Malawi were ST313 ($n = 549$) and ST19 ($n = 34$). A small number of isolates ($n = 25$) were typed as ST313 single-locus variants (ST3257 ($n = 17$), ST2080 ($n = 3$), ST302 ($n = 3$), ST4200 ($n = 1$) and ST4274 ($n = 1$)). The contextual dataset contained two sequence types, ST313 ($n = 43$) and ST19 ($n = 29$). In sum, ST313 isolates originated from nine African countries (Fig. 1). Previous genome-based analyses of the African ST313 epidemic included isolates from eastern and central Africa[6,11]. Here, by including several Western Africa countries, we have expanded the known geographic range of ST313[6,8,11,20–23] to include Cameroon, Central African Republic, Niger, Senegal, Sudan and Togo.

To investigate population structure, a core genome maximum-likelihood phylogeny was constructed, and cluster designation was performed. Three major ST313 clusters, four ST19 clusters and a small number of miscellaneous strains were identified (Extended Data Fig. 1). Multi-locus sequence type variants formed discrete sublineages. The three ST313 clusters were clonal, with cluster 1 and 2 corresponding to previously defined lineages (ST313 L1 and L2)[5].

The genomes of contemporary isolates revealed that a third lineage has been circulating in Malawi since 2016 (ST313 L3; Fig. 2). Characterization based on long-read sequencing showed that ST313 L3 strain BKQZM9 had a circular chromosome (4.8 Mb) with 99.98% sequence identity to the ST313 L2 reference strain D23580[5].

To fully contextualize ST313 L3, we constructed a core gene maximum-likelihood phylogeny, including previously published ST313 genomes (Supplementary Table 3), alongside the current dataset (Extended Data Fig. 2). ST313 L3 forms a monophyletic cluster within a group of ST313 strains isolated in the United Kingdom and Brazil[7,8], raising the possibility of an international transmission event. ST313 L3 isolates are closely related, with a maximum pairwise distance of 16 SNPs. To determine whether ST313 L3 represents an outbreak in Malawi, it would be necessary to combine genome-based information with details on time, person and place. Patient-level GPS data were unavailable, therefore the geospatial distribution of ST313 L3 is currently unknown.

**Accessory genome of *S.* Typhimurium ST313 lineages.** To understand the relationship between ST313 L3 and other ST313 lineages, we examined the accessory genome in the context of population structure (Fig. 2). All lineages carried the virulence plasmid pSLT[24], and ST313 L2 additionally contained plasmids pBT1, pBT2 and pBT3, consistent with previous studies[5,17]. ST313 L1 carried a previously unreported 8,274-base pair (bp) plasmid, sharing 99.27% sequence identity with pAnkS[25] (GenBank NC_010896.1), which encodes the LsoA–LsoB toxin–antitoxin proteins. This toxin–antitoxin system functions as a bacteriophage exclusion system in *Salmonella*[26] (Supplementary Discussion 1). ST313 L3 strain BKQZM9 contained the 95 kb plasmid (pSLT), a 1,975-bp plasmid (pBT3) and a 2,556-bp plasmid (pBT2, identified using short-read data only). Thus, the plasmid profile of ST313 L3 shared more similarity with ST313 L2 than other lineages, but lacked pBT1 (Fig. 2, S1).

A pairwise comparison of reference strains BKQZM9 (ST313 L3), D23580 (ST313 L2) and U2 (the representative UK-isolated ST313 strain) (Extended Data Fig. 3) revealed chromosomal differences involving the prophage repertoires (Fig. 2 and Supplementary Information). The common *S.* Typhimurium prophages Gifsy-1, Gifsy-2[27] and ST64B[28] were identified in all three ST313 African lineage reference strains. ST313 L3 carried a Fels-2-like prophage that was absent from the ST313 L2 and the UK-isolated ST313 representative, and shared 99.99% sequence identity to the prophage RE-2010[29] found in other extraintestinal *Salmonella* serovars, including a global outlier cluster of *Salmonella* Enteritidis[30] and *Salmonella* Panama[31,32] (Extended Data Fig. 3). Prophage BTP5 was absent from the ST313 L3 reference strain. BTP5 is associated with ST313 L1 and L2[33], but is generally absent from UK-isolated ST313[20].

Prophage BTP1 was not found in ST313 L3, although a related P22-like prophage with 74% fragmented sequence identity to BTP1 occupied the same attachment site (Extended Data Fig. 4a). The P22-like prophage had almost identical replication, lysis and capsid genes to BTP1, differing only in the Ea, immunity, tail and *gtr* regions, and lacking the *bstA*[34] gene. The ST313 L3 P22-like prophage carried the *sieB* gene, which encodes a superinfection exclusion phage immunity protein[35]. The SieB proteins of ST313 L3 and P22 were 92% identical, with the two proteins sharing 177 out of 192 amino acids. Carriage of a gene that mediates abortive phage infection could confer a selective advantage to ST313 L3, reflecting the functional role of the BTP1 prophage-encoded BstA virus defence protein in ST313 L2[34]. BstA defends bacterial cells that carry BTP1 against exogenous attack by a variety of lytic phages, conferring a beneficial trait to ST313 L2 without sacrificing lytic autonomy[34]. Although BTP1 is absent from the UK reference strain U2, two UK-isolated ST313 strains (U15 and U8[7]) carried a P22-like prophage with 99.96% sequence identity to the one found in ST313 L3 (Extended Data Fig. 4b). In summary, the prophage repertoire of ST313 L3 shares similarities with UK-isolated ST313.

---

**Fig. 1 | Bloodstream isolates of *Salmonella* Typhimurium used in this study.** Isolates were collected from either the Unité des Bactéries Pathogènes Entériques of the Institut Pasteur Centres ($n = 72$, light blue) or the MLW Clinical Research Center ($n = 608$, dark blue). Bar graphs show the number of isolates of different sequence types collected by centre per year. Donut charts indicate the proportion of sequence types collected per country and show the total number of isolates from each location in the centre. Note that only isolates from African countries (M'gascar, Madagascar; Congo, Republic of the Congo) are shown in donut plots. Letters in superscript relate to the location on the map. The colour code is shown at the bottom of the figure.

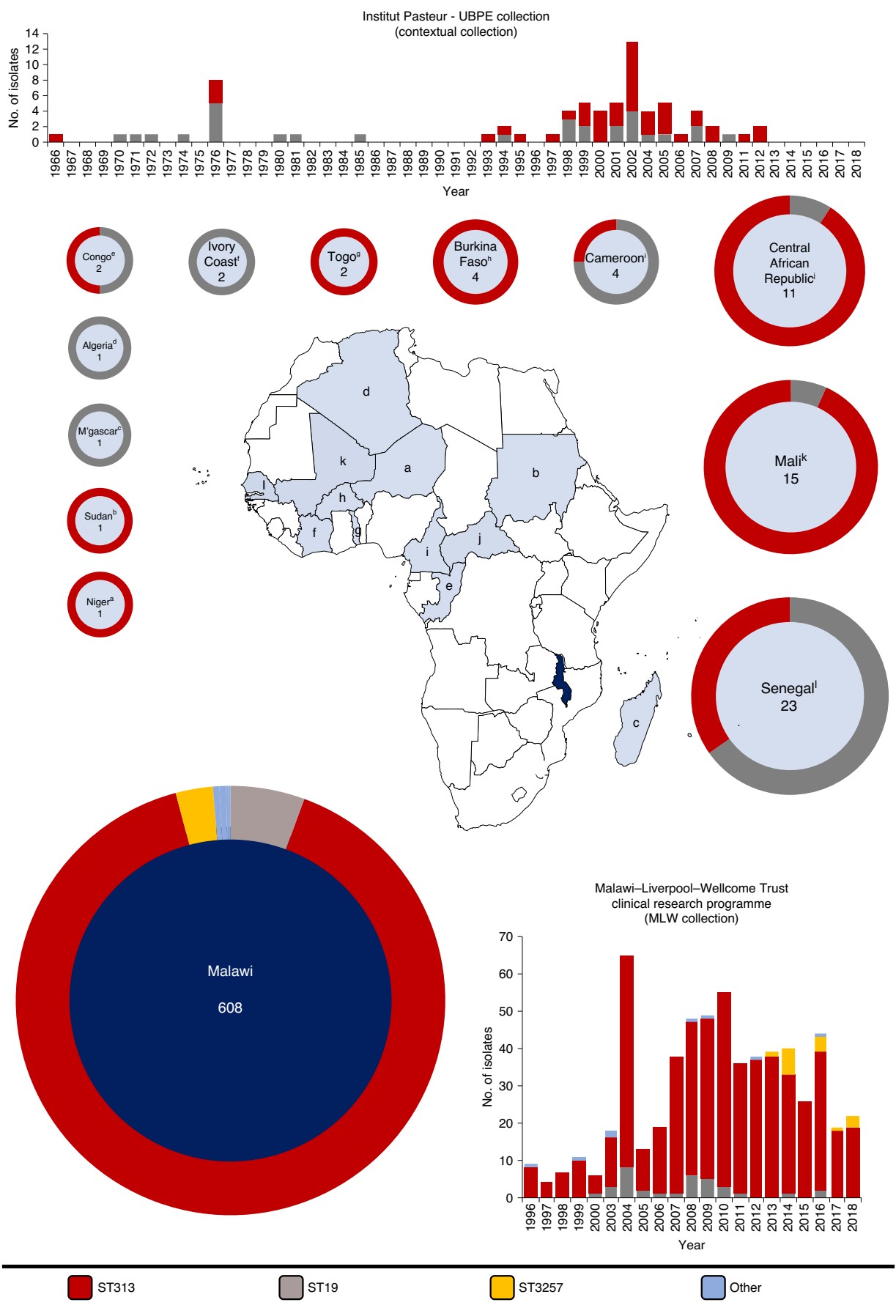

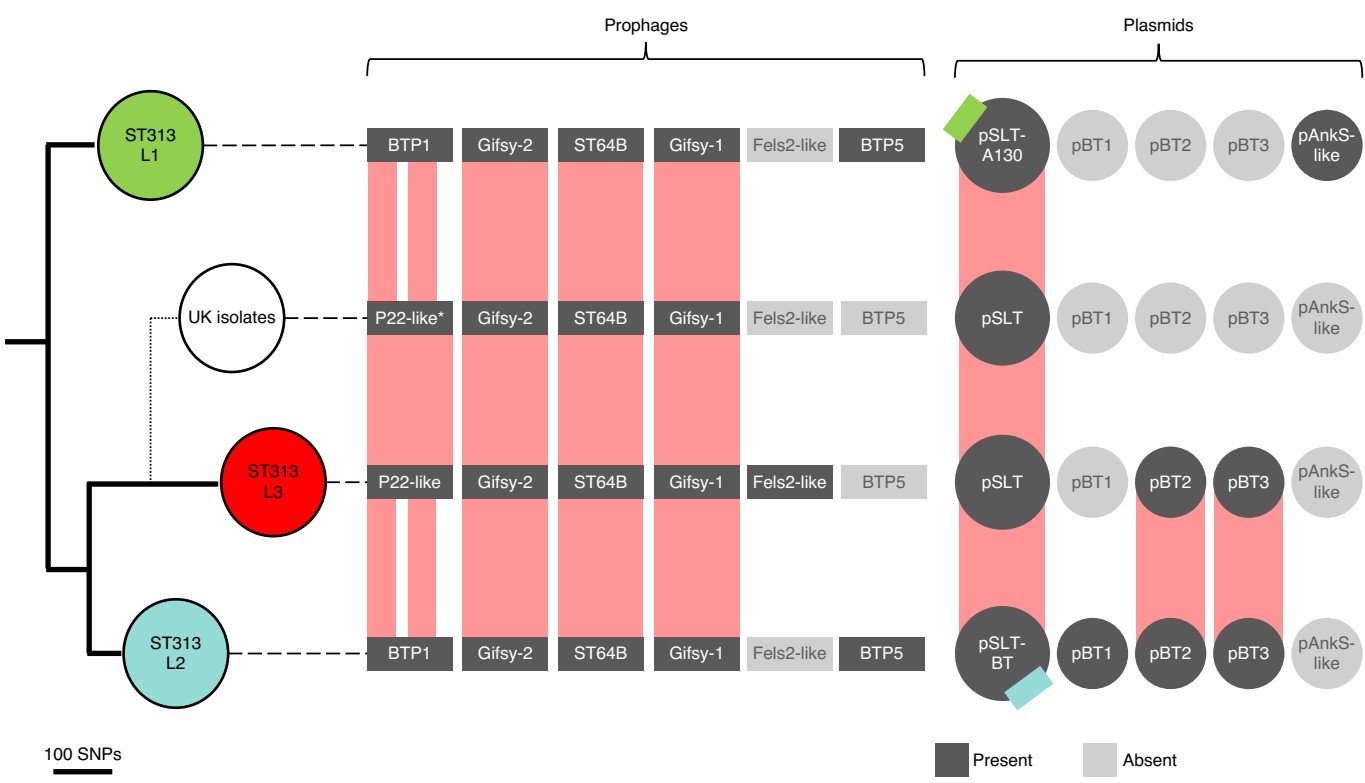

**Fig. 2 | The four major ST313 clusters have different prophage and plasmid repertoires.** Maximum-likelihood phylogeny demonstrating the population structure of ST313 L1, L2 and L3. Approximate location of UK lineages is indicated schematically (Extended Data Fig. 2) due to the diversity of isolates. Prophage and plasmid repertoire of the reference strain for each lineage are shown (dark grey indicates presence; light grey represents absence). Reference isolates used were A130 (ST313 L1), D23580 (ST313 L2), BKQZM9 (ST313 L3) and U2 (UK strains). Red blocks represent the extent of conservation between lineage reference genomes, with white gaps indicating missing regions. Coloured squares on the pSLT plasmid represents different AMR cassettes. *P22-like prophage is absent in UK reference (U2), but present in some UK strains.

**Genomic epidemiology of AMR.** To obtain epidemiological insights into the emergence of ST313 lineages, we investigated the dynamics of AMR. The draft genome sequences of all 680 isolates were examined for genes and mutations that confer reduced susceptibility to antimicrobials (Supplementary Discussion 2). In total, 65% (n = 440) of isolates had an MDR genotype[36]. Among ST19 isolates, only 11% (7 out of 63) of isolates were resistant to at least 1 antimicrobial. However, among ST313 isolates, 94% of isolates were resistant to at least 1 antimicrobial. The number of chloramphenicol-resistant isolates of ST313 L2 have decreased over time, with 89.19% (n = 66 out of 74) before 2005, decreasing to 84.11% (n = 307 out of 365) between 2006 and 2015 and 65.28% (n = 47 out of 72) between 2016 and 2018 (Fig. 3 and Supplementary Table 2). Interestingly, there have been changes in antimicrobial usage policy at the local level in Malawi during this timeframe, including the phased withdrawal of chloramphenicol from clinical practice beginning in 2001.

Overall, ST313 in this study had numerous genetic determinants of antibiotic resistance, with 29 different AMR genotype patterns involving 11 antimicrobials (Fig. 3). The AMR phenotype of ST313 L1 and L2 is encoded by resistance cassettes carried on a composite Tn21-like transposable element, inserted in the S. Typhimurium virulence plasmid pSLT[5–7,14]. The plasmid backbone of pSLT was present in all ST313 strains and we observed lineage-specific variation in the Tn21 insertion site, consistent with previous reports[5]. The Tn21-like region varied between isolates, reflecting the AMR profile (Extended Data Fig. 5). Plasmid pAnkS[ST313-L1] carried the $bla_{TEM}$ gene which mediated beta-lactam resistance of ST313 L1.

ST313 L3 was pan-susceptible to all antimicrobials tested, both genotypically and phenotypically (Fig. 2), as were the UK-isolated ST313[20]. In contrast to L1 and L2, ST313 L3 did not carry an MDR cassette on the pSLT-BT plasmid (Extended Data Fig. 6). Phylogenetic reconstruction of the pSLT plasmid (Extended Data Fig. 7) reflected the core genome SNP-based phylogeny, demonstrating four phylogenetic clades (ST313 L1, L2, L3 and ST19). There was no evidence of a scar associated with excision of the Tn21-like transposable element around location 48,530 on the ST313 L3 pSLT plasmid.

Less than 1% (n = 5/180) of African S. Typhimurium ST313 L2 carried mutations associated with reduced fluoroquinolone susceptibility (for example, gyrA[37]), originating in Malawi (n = 3), Mali (n = 1) or Cameroon (n = 1). The GyrA proteins had either S83F, D87N or D87Y amino acid substitutions. All five strains were phenotypically susceptible to ciprofloxacin. Ongoing surveillance will be important for identification of strains with triple mutations which would confer full resistance to this clinically important antibiotic.

**The evolutionary trajectory of S. Typhimurium ST313.** Dates for lineage divergence were provided by Bayesian phylogenetic inference (Fig. 4 and Extended Data Fig. 8). The most-recent common ancestor (MRCA) of all ST19 and ST313 sampled in this study was from around AD 918 (95% highest posterior distribution (HPD) AD 570–1080) and the MRCA of all ST313 sampled in this study (Fig. 4, event 0) was from around AD 1566 (95% AD HPD 1271–1953). The incorporation of historical and diverse isolates that predate those used in previous phylodynamic analysis[6,7] generates an older MRCA of ST313 lineages than previously described[7]

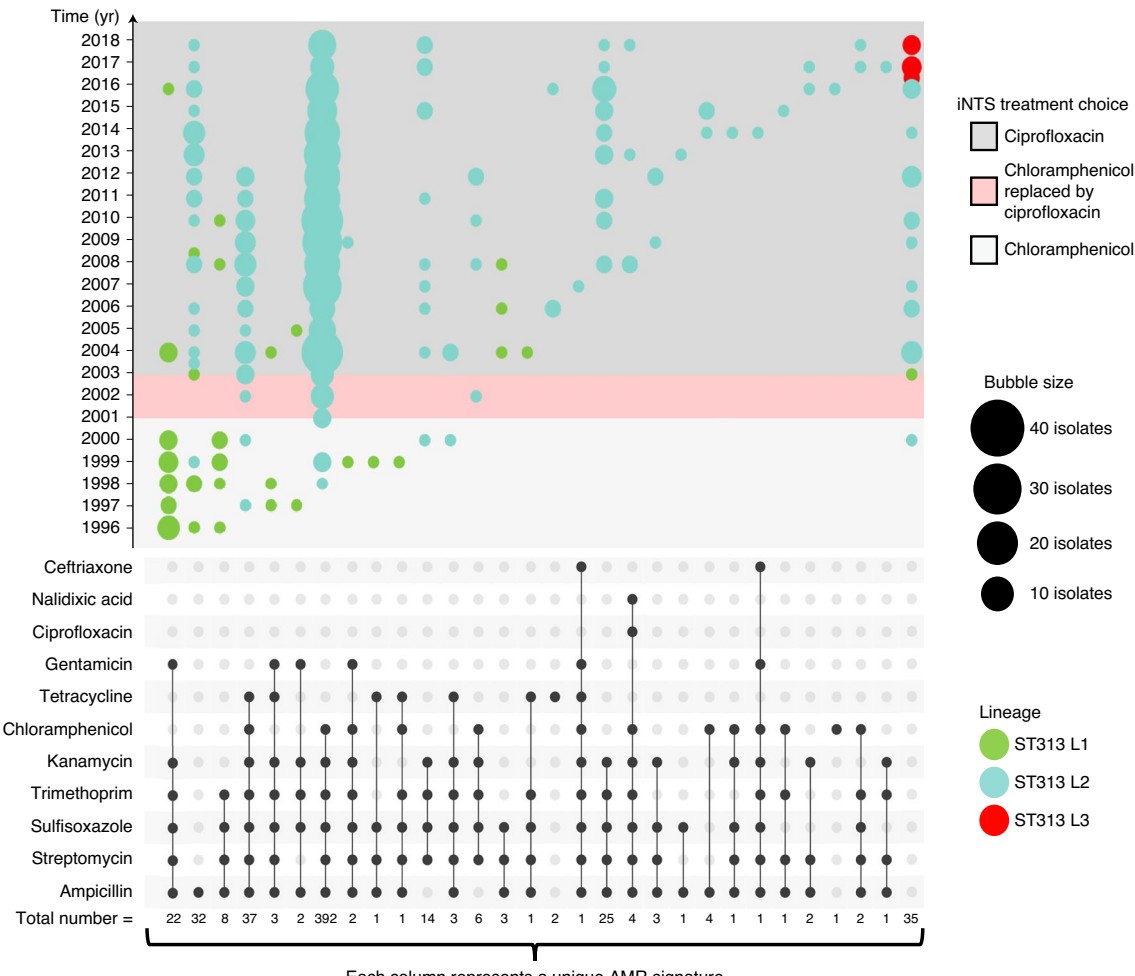

**Fig. 3 | Temporal AMR trends in *S*. Typhimurium lineages (1996–2018).** The combination matrix (bottom) depicts predicted AMR patterns of *S*. Typhimurium. Isolates collected before 1996 were sampled only sporadically, and thus were excluded from analysis. Within the combination matrix, dark grey circles indicate genome-predicted resistance and the vertical combination of grey circles represents the resistance profile. Total number of isolates with each resistance profile is indicated below the matrix. The combination matrix was created using the UpSetR package[87]. The bubble plot (top) depicts the relative number of isolates (bubble size) with each resistance profile (combination matrix) per year (*y* axis). Lineage assignments are shown using bubble colour to identify lineage-specific AMR trends. Local Malawi antimicrobial usage policy[9] is highlighted in background colours overlaid on the timeline in the bubble plot. The bubble plot was created using R ggplot2[88].

(around AD 1787). However, we note that the branch linked to the MRCA is deeply rooted in the phylogeny and the 95% HPD values reported here have a substantial range. A limitation of Bayesian analysis is that it relies upon a series of user-defined prior probability distributions that influence the estimation of tip dates (Methods).

The MRCA of ST313 L1 dated to about AD 1794 (95% HPD AD 1738–1965), with Malawian ST313 L1 forming a discrete sublineage with an MRCA dating to around AD 1950 (95% HPD AD 1921–1986) (Fig. 4, event 1). The ST313 L2 MRCA (Fig. 4, event 2) dated to around AD 1948 (95% HPD AD 1929–1959). These MRCA estimates overlap with previous 95% HPD values for ST313 lineage dating[6]. The MRCA of ST313 L3 (Fig. 4, event 3) was AD 2007 (95% HPD AD 1998–2012), showing that the clonal expansion of ST313 L3 occurred in recent evolutionary history.

**Phylodynamics of pseudogene formation in *S*. Typhimurium ST313.** To explore the association of pseudogenization and invasive salmonellosis, we characterized the stepwise progression of functional gene loss associated with host adaptation that probably facilitated the ST313 epidemic in Africa (Methods). This phenomenon is exemplified by the timeline of pseudogene accumulation in ST313 L2 shown in Fig. 4.

Our discovery of ST313 L3 allowed the timeline of gene-degradation events to be scrutinized in more detail. For example, mutations in *melR* (F311L), *flhA* (A166T) and *pipD* (283-bp deletion) occurred prior to the divergence of ST313 L3 and ST313 L2, while the mutations in *bcsG* (W247*), *sseI* (IS*26* insertion) and *lpxO* (E198*) preceded the clonal expansion of ST313 L2. A small number of gene-degradation events were associated only with ST313 L3, specifically an additional non-synonymous mutation in *ratB* (G820S) and a 109-bp deletion in *pipD*. Both mutations occurred in genes that had been pseudogenized earlier in evolutionary history, and so are unlikely to cause specific phenotypic changes. The *macB*, *sseI* and *lpxO* genes that were non-functional in ST313 L2 were functional in ST313 L3, as reported previously for the UK ST313 representative (U2)[7]. We recently reported the functional significance of the pseudogenization of *macB* for ST313 L2[18]. The frameshift event that inactivated the *macB* gene occurred between AD 1948 and AD 1953 (95% HPD AD 1931–1963), representing the final stage in the evolution of ST313 L2 detected in our analysis (Fig. 4).

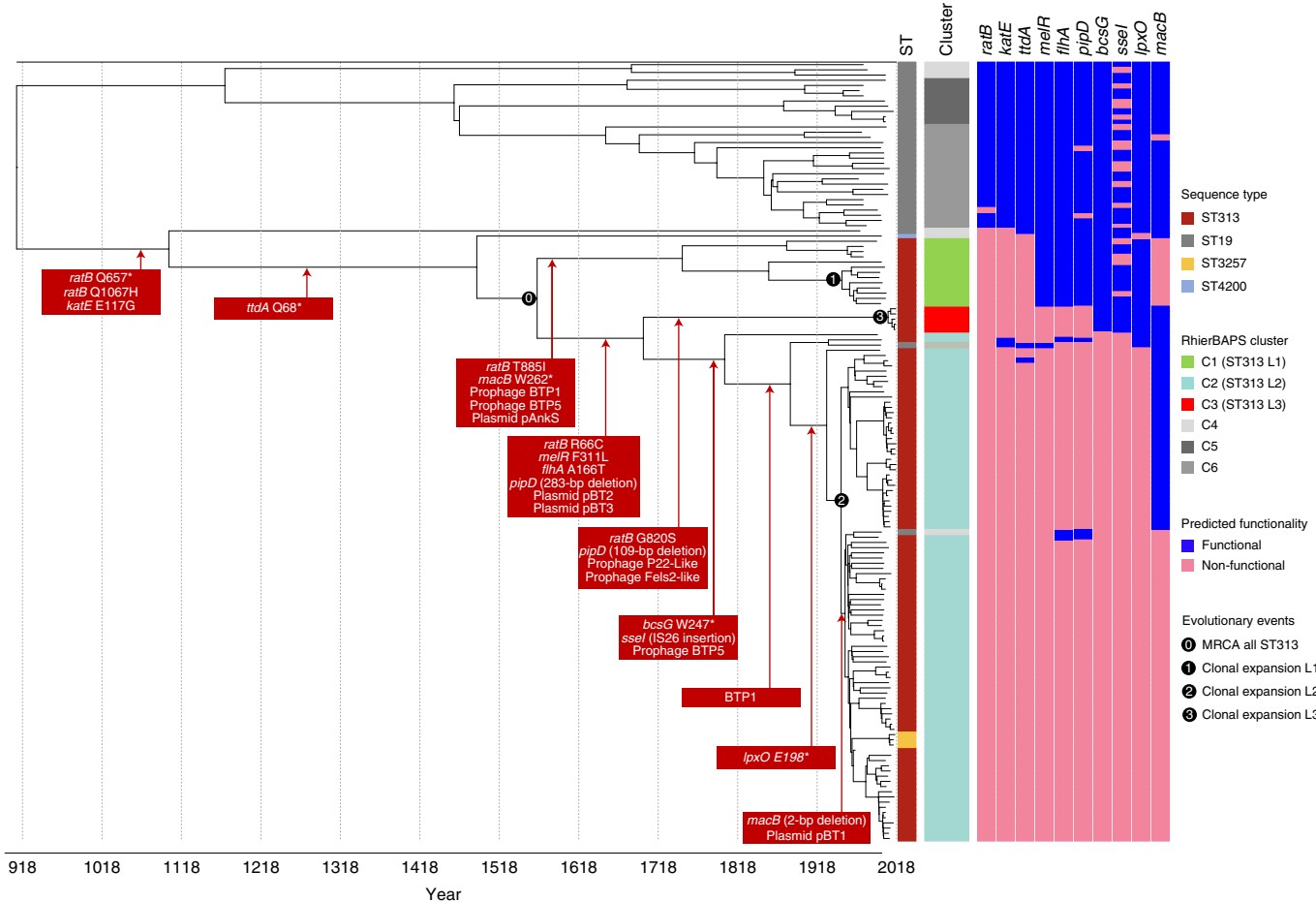

**Fig. 4 | Stepwise evolution of *S*. Typhimurium responsible for BSI in Africa.** Chronograph of 150 *S*. Typhimurium strains isolated from the bloodstream of human patients with iNTS disease. The choice of the 150 isolates from both the Malawian and contextual datasets is described in Methods. The figure shows a maximum clade credibility tree. Adjacent colour strips are as follows (from left to right); ST, lineage assignment (rHierBAPs) with the three major ST313 lineages highlighted in colour. Predicted functionality (Methods) is depicted as a colour strip for each gene and is based on whole-genome-based predictions of SNPs likely to play a functional role. Genome-degradation events that generate functionally relevant pseudogenes are displayed in red boxes overlying the chronograph. An asterisk represents a premature stop codon. The numbers in black circles indicate four key evolutionary events. Figure visualized using iTOL[69].

**Function of evolutionarily important *S*. Typhimurium ST313 genes.** To complement our genome-derived evolutionary insights, we linked a number of lineage-specific phenotypes to individual SNPs (summarized in Table 1 and detailed fully in Supplementary Table 4). For example, the KatE catalase protects bacteria from environmental oxidative stress, and shows a maximal activity in stationary phase cultures of *S*. Typhimurium ST19[15]. The E117G KatE mutation caused low level catalase activity in ST313 L1, L2, L2.1 and the UK-isolated ST313[7,11,15]. We report a similarly low activity in ST313 L3 (Table 1 and Supplementary Table 4). Catalase is required for biofilm formation[38], a phenotype associated with survival outside the host, leading to the suggestion that pseudogenization of *katE* in ST313 reflects adaptation to a restricted host range[16].

The red, dry and rough (RDAR)-negative phenotype of ST313 L2[39] is caused by pseudogenization of *bcsG*[15] and two nucleotide changes in the *csgD* gene promoter region[39]. ST313 L3 has an intermediate RDAR phenotype, with an incomplete wrinkling pattern (Supplementary Table 4). ST313 L3 has a functional *bcsG* gene and had the same *csgD*-associated −189 promoter mutation as ST313 L1 and L2[39]. Our finding that ST313 L3 has lost traits required for stress resistance and biofilm formation is consistent with a reduced

requirement for environmental survival and the mounting evidence for human-to-human transmission of ST313[40,41].

*S*. Typhimurium ST19 relies on carbon metabolism to colonise the mammalian gastrointestinal tract[42], whereas genes required for utilization of specific carbon sources, such as tartrate and melibiose, are not functional in ST313 lineages[17]. ST313 L3 was also unable to grow with tartrate or melibiose as a sole carbon source, consistent with the pseudogenization of the *ttdA* and *melR* genes in both ST313 L3 and ST313 L2[5,17] (Fig. 4).

Further evidence of genome degradation is provided by the pseudogenization of the genes *ratB* in all ST313 lineages, *pipD* in ST313 L2 and L3, and *lpxO* in ST313 L2. The PipD effector protein has been implicated in gastrointestinal pathogenesis of ST19, although a causal relationship has not been demonstrated[43,44]. The *lpxO* gene is pseudogenized by a stop codon in ST313 L2[5]. LpxO hydroxylates lipid A, a modification required for virulence of *S*. Typhimurium ST19[45]. We used mass spectrometry to show that the lack of functional LpxO caused structural modifications of lipid A in ST313 L2 (Table 1, Supplementary Table 4). The functionality of the *lpxO* gene in ST313 L3, reflects the distinct evolutionary path of this lineage.

**Table 1 | Summary of the phenotypic impact of pseudogenes in ST313 lineages**

| Gene Name | Description | Phenotype |
|---|---|---|
| *ratB* (STM2514) | Secreted outer membrane protein | Inactivation of *ratB* is associated with gut persistence in mouse model and is likely to have reduced the enteric potential of ST313 (ref. [14]). |
| *katE* (STM1318) | Stationary phase catalase | Catalase activity is reduced across all ST313 L1 and L2 isolates[7,15]. We confirmed that ST313 L3 has a lower catalase activity than ST19. |
| *ttdA* (STM3355) | ʟ(+)-tartrate dehydratase | ʟ-tartaric acid and dihydroxyacetone cannot be used as sole carbon sources by ST313 L1 or L2 (ref. [14]). We confirmed experimentally that ST313 L3 is unable to grow on ʟ-tartaric acid as a sole carbon source. |
| *melR* (STM4297) | Melibiose operon response regulator | ST313 L2 cannot grow on melibiose as a sole carbon source[17,82]. We confirmed experimentally that ST313 L3 was unable to grow on melibiose as a sole carbon source. |
| *flhA* (STM1913) | Flagella biosynthesis protein | Comparison of the ST313 L2 D23580 *flhA*[474] mutant to wild type shows that the A166T SNP carried by ST313 L2 and ST313 L3 causes reduced motility[17]. |
| *pipD* (STM1094) | Pathogenicity island encoded protein D | Although a causal relationship is unproven, the PipD effector protein of ST19 contributes to macrophage persistence in murine models[14,17,43]. |
| *bcsG* (STM3624) | Cellulose biosynthetic enzyme | Biofilm formation is impaired in ST313 L2 due to a mutation in *bcsG*[15]. Here we show that biofilm formation in ST313 L3 is also impaired. The genetic basis is unknown. |
| *sseI* (STM1051) | Type III secretion effector protein (SPI-2) | Studies in mice demonstrate that an accumulation of SNPs within the *sseI* gene of ST313 L2 increases the ability of the bacteria to disseminate rapidly from the gut to the draining lymph nodes[16]. |
| *lpxO* (STM4286) | Putative membrane-bound β-hydroxylase | We compared the lipid A of ST313 L2 (D23580) and ST19 (4/74) during growth in SPI-2-inducing media. By mass spectrometry, we confirmed that the lipid A of ST313 L2 lacks the LpxO-mediated modification associated with ST19. |
| *macB* (STM0942) | Putative ABC transport protein | MacAB is involved in oxidative stress resistance[13] and antibiotic resistance to macrolides[81]. In ST313 L1 and L2, variants of the MacAB–TolC tripartite efflux pump affect replication in macrophages and influence fitness during colonization of the murine gastrointestinal tract[18]. |

Associated phenotypic data are presented in Supplementary Table 4.

**Invasiveness potential of ST313 L3.** To determine the potential of ST313 L3 to cause BSI, the invasiveness index of each sample was calculated and compared between different ST313 lineages using the Wilcoxon–Mann–Whitney test[46]. The 'invasiveness index' represents the extent of genome degradation and diversifying selection specific to invasive serovars, using a proven set of 196 extraintestinal predictor genes[47]. Consistent with previous studies, we observed a significant increase in the invasiveness index of ST313 L2 compared with ST313 L1 ($W = 7,636.5$, $P < 0.001$), and ST313 L1 compared with ST19 ($W = 552.5$, $P < 0.001$)[11,47]. ST313 L3 had a particularly high invasiveness index (median = 0.187, s.d. = 0.008), which was significantly greater than that of ST19 (median = 0.110, s.d. = 0.017), ST313 L1 (median = 0.129, s.d. = 0.008) or ST313 L2 (median = 0.134, s.d. = 0.006) ($W = 0$, $P < 0.001$) (Fig. 5). ST313 L3 also had a significantly greater invasiveness index than the UK-isolated ST313 (median = 0.134, s.d. = 0.018), ($W = 480$, $P < 0.001$).

In total, 17 of the 196 extraintestinal predictor genes had undergone additional degradation in ST313 L3 (isolate BKQZM9) compared with ST313 L2 (D23580), involving the presence of non-synonymous SNPs, indels or gene loss (Supplementary Table 5). A number of the degraded genes in ST313 L3 are required for colonisation of the gastrointestinal tract by *S.* Typhimurium ST19, including *mrcB*, a gene that allows *Salmonella* Typhi to survive in the presence of bile[48]. In *S.* Typhimurium, a functional *damX* gene is required for resistance to bile[49], and was degraded in seven of the nine ST313 L3 isolates. The *napA* gene had extensive deletions in all but one of the ST313 L3 isolates, and encodes a periplasmic

nitrate reductase required for gut colonisation[17,50]. Together, our findings suggest that ST313 L3 is in the process of adapting from an intestinal to a systemic lifestyle.

Clearly, this in silico approach does not prove that ST313 L3 has adapted to an extraintestinal lifestyle. In the past, infection models have provided understanding of the virulence and systemic spread of *Salmonella* pathovariants that cause gastroenteritis[51]. However, cellular and animal infection models have failed to discriminate between levels of invasiveness of ST313 L1 and L2[14] and an ST313 sublineage identified in the Democratic Republic of Congo[11]. Although the invasiveness index cannot yet be experimentally validated, *Salmonella* isolates with different invasiveness indices produce distinct clinical symptoms in a human population[52]. The development and validation of experimental approaches for the robust measurement of the invasiveness of African *S.* Typhimurium will be an important focus for future research, but goes beyond the scope of this paper. Meanwhile, the 'invasiveness index' is a useful tool for monitoring the genetic signatures associated with invasiveness and host adaptation in ST313 lineages.

## Perspective

Our data provide an expanded and contemporary insight into the dynamics of *S.* Typhimurium-associated BSI in Africa, elucidating the stepwise evolution of ST313. This study provides a snapshot of the large repertoire of genomic changes that shaped the emergence of successful ST313 lineages. The stepwise pseudogenization of genes reported here is consistent with the host adaptation of *S.* Typhimurium ST313 lineages over evolutionary history[6,40] and

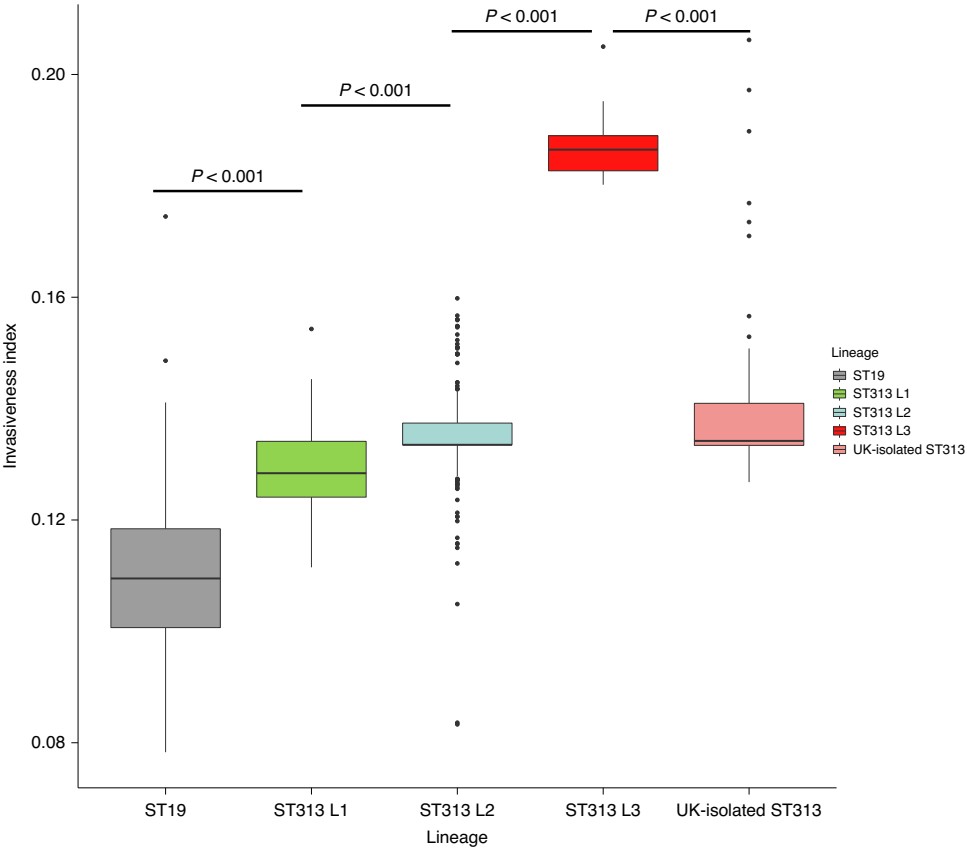

**Fig. 5 | Invasiveness index of ST19 and ST313 lineages.** Box plot representing the distribution of invasiveness index values for all genome sequences included in this study summarized by lineage assignment. UK-isolated ST313 refers to those described in ref. [7]. Groups were compared using a two-sided Wilcoxon–Mann–Whitney test[46] and the resultant *P*-values were all less than 0.001. Number of isolates in each group: ST19 (*n* = 66), ST313 L1 (*n* = 52), ST313 L2 (*n* = 550), ST313 L3 (*n* = 9) and UK-isolated ST313 (*n* = 59). In the box plot, centre lines represent median values, box limits represent upper and lower quartiles, whiskers represent 1.5× the interquartile range and individual points represent outliers. The box plot was created using R ggplot2[88].

supports mounting evidence for a human ST313 reservoir[40]. Our findings reflect similar patterns of evolution found in other *Salmonella* serovars that currently cause BSI disease globally. For example, a third of all iNTS disease cases in Africa are caused by *S.* Enteritidis which also shows signs of genome degradation consistent with niche adaptation[30,53].

We discovered that ST313 L3 emerged in 2016 as a cause of BSI in the Malawian population, and had an MRCA from around AD 2007. It is possible that the phased removal of chloramphenicol from clinical practice following local policy changes in Malawi between 2002 and 2005 created a window of opportunity for the emergence of fully susceptible ST313 L3 (Supplementary Discussion 2). ST313 L3 forms a monophyletic cluster within a diverse group of ST313 originally identified in the UK[7]. The combination of our phylogenetic and accessory genome analyses led us to conclude that ST313 L3 has been introduced into the Malawian population, probably via an international transmission event. Importantly, ST313 L3 had an elevated invasiveness index compared to ST313 L2, unlike the UK-isolated ST313 strains that had a lower invasiveness index than ST313 L2[47]. The majority of genes unique to ST313 L3 were plasmid- or prophage-encoded, including by the prophage RE-2010[29] which is found in other extraintestinal *Salmonella* pathovariants[30–32]. Apart from accessory genome composition, other ST313 L3-specific changes involve loss-of-function SNPs in genes that are not required for systemic infection. The findings predict that ST313 L3 will be better adapted to cause extraintestinal infection than other ST313

lineages from both the UK and Malawi, possibly explaining the emergence of ST313 L3.

## Methods

**Dataset.** *Salmonella* samples were derived from two archived blood culture collections. The main dataset was sourced from the MLW in Blantyre, Malawi and consists of over 14,000 *Salmonella* bloodstream isolates from patients with iNTS disease at the Queen Elizabeth Central Hospital between 1996 and 2018. Isolates listed as *S.* Typhimurium were extracted from the metadata file for the entire collection and stratified by AMR profile into the following four categories: susceptible, resistant to one first-line agent, resistant to two first-line agents, or MDR. First-line agents were considered to be ampicillin, cotrimoxazole and chloramphenicol. Random subsampling was then performed using a random number generator (Excel, Microsoft) and selected samples were collected and resuscitated from the freezer archives at MLW. We successfully recovered 647 isolates for whole-genome sequencing.

To complete the MLW data, a contextual dataset was sourced from the Institut Pasteur, Paris. The dataset consisted of 96 *S.* Typhimurium isolates collected from human extraintestinal sites from patients contaminated in Algeria, Burkina Faso, Cameroon, Central African Republic, Republic of the Congo, France, Côte d'Ivoire, Madagascar, Mali, Niger, Senegal, Sudan, Togo and Vietnam between 1966 and 2012. All 96 *S.* Typhimurium samples were selected and sent for whole-genome sequencing.

**Whole-genome sequencing of short reads.** Isolates were prepared by inoculating a single bead of frozen *Salmonella* stock into a FluidX 2D Sequencing Tube (FluidX) containing 100 μl buffered peptone water (Oxoid, CM0509) and incubating overnight at 37 °C. Thermolysates were generated by heat-killing cultures in a 95 °C water bath for 20 min before DNA extraction (MagAttract kit, Qiagen) and whole-genome sequencing, as part of the 10,000 *Salmonella* Genomes Project[54]

(https://10k-salmonella-genomes.com/). Illumina Nextera XT DNA Libraries were prepared (Illumina, FC-131-1096) and sequenced (Illumina HiSeq 4000) in multiplex (768) as 150-bp paired-end reads.

**Assembly and annotation of short reads.** Trimmomatic[55] v0.36 was used to trim adapters and Seqtk v1.2-r94 (https://github.com/lh3/seqtk) was used to trim low-quality regions using the trimfq flag. Fastqc v0.11.5 (https://www.bioinformatics.babraham.ac.uk/projects/fastqc/) and multiqc v1.0 (http://multiqc.info) were used to pass sequence reads according to the following criteria: passed basic quality statistics, per base sequence quality, per base N content, adapter content and an average GC content of between 47% and 57%. Only high-quality reads were used in downstream analysis. Unicycler[56] v0.3.0 was used to assemble genomes and QUAST[57] v4.6.3 was used to evaluate assembly quality to standards consistent with Enterobase[58]. Specifically, N50 > 20 kb, 600 or fewer contiguous sequences, total number of bases between 4 Mbp and 5.8 Mbp. Prokka[59] v1.12 was used to annotate the genomes.

**Sequence typing.** Serotyping was confirmed using the *Salmonella* In Silico Typing Resource[60] v1.0.2 and compared with the original metadata table. The strains were assigned a multi-locus sequence type (MLST) using MLST[61] v2.10 on the basis of the conservation of seven housekeeping genes.

**Reference mapping to D23580.** Trimmed sequencing data were mapped against the *S.* Typhimurium reference genome D23580 using BWA mem[62] v0.7.10-r789. The reference genome consists of the D23550 chromosome (GenBank accession: LS997973.1), and four plasmids including pSLT-BT (accession: LS997974.1), pBT1 (accession: LS997975.1), pBT2 (accession: LS997976.1) and pBT3 (accession: LS997977.1). Mapped reads were then cleaned and sorted using the SAMtools suite[63] v1.7. Reads were realigned against the reference using GATK[64] v3.7 by creating targets for realignment (RealignerTargetCreator) and performing realignment (IndelRealigner). Removal of optical duplicates was completed using Picard v2.10.1-SNAPSHOT (https://broadinstitute.github.io/picard/). Sequence variants were called using Bcftools v1.9-80 (http://samtools.github.io/bcftools) to generate a reference-based pseudogenome for each sample with greater than 10× depth. QualiMap[65] v2.0 was used to identify a mean sample depth of 35.17× across all isolates.

**Phylogenetic reconstruction of dataset.** High-quality pseudogenomes (MLW *n* = 608, contextual *n* = 72) were concatenated, plasmids were excluded and Gubbins[66] v2.2 was used to remove recombinant regions and invariable sites. The resultant multiple sequence alignment of reference-based pseudogenomes (12,013 variant sites) was used to infer a maximum-likelihood phylogeny using RAxML-ng[67] v0.6.0 with 100 bootstrap replicates to assess support. To assign clusters, rhierBAPs[68] was used, specifying two cluster levels, 20 initial clusters and infinite extra rounds. Visualizations were performed using interactive Tree Of Life (iTOL)[69] v4.2.

**Phylogenetic reconstruction of dataset and publicly available dataset.** For contextual interpretation, previously published *S.* Typhimurium ST313 sequences were retrieved from public repositories (Supplementary Table 2) and constructed in a core gene phylogeny with our data. Roary[70] v3.11.0 was used to generate a core gene alignment and SNP-sites[71] v2.3.3 was used to extract SNPs. The resultant multiple sequence alignment (15,240 variant sites) was used to construct a maximum-likelihood phylogeny using RAxML-ng[67] v0.6.0 with 100 bootstrap replicates to assess branch support. Visualizations were made using iTOL[69] v4.2.

**Temporal phylogenetic reconstruction.** To determine the evolutionary history of ST313, a chronogram was produced using Bayesian phylogenetic inference. Because the MLW collection is substantially larger than the contextual collection and to reduce computation time, 150/680 samples were chosen for inclusion in the analysis as described below. Sampling reflected original sample selection for this study and included all contextual isolates which passed quality control (*n* = 72) and a subset of MLW isolates randomly selected using a random number generator (*n* = 78, Microsoft Excel). A reference-mapped multiple sequence alignment (7,231 variant sites) was created as described above. BEAUTI[72] v2.6.1 and BEAST2[72] v2.6.1 were used to create and execute three independent chains of length 250,000,000 with 10% burn in, logging every 1,000 and accounting for invariant sites. We included the prior assumptions of a coalescent Bayesian skyline model for population growth, and a relaxed log normal clock rate to account for rate heterogeneity amongst branches (the full model is described in Supplementary Data 1). Tracer[73] v1.7.1 was used to assess convergence, with all parameter effective sampling sizes being >200. LogCombiner[72] v2.6.1 was used to combine tree files and DensiTree v2.2.7[74] was used for visualization. Finally, a maximum clade credibility tree was created using TreeAnnotator[72] v2.6.0.

**Determination of invasiveness index.** The invasiveness index of each isolate was calculated using previously defined methods[47]. Because isolates used in our study originated from human bloodstream, we used an invasiveness index model that was pre-trained using a mixture of gastrointestinal and extraintestinal salmonellae[47]. Specifically, samples were analysed using 196 top predictor genes

for measuring invasiveness in *S. enterica*. The machine-learning approach uses a random forest classifier and delta-bitscore functional variant-calling to discriminate between the genomes of invasive and gastrointestinal *Salmonella*. The invasiveness index has been validated using multiple *Salmonella* serovars, and the approach clearly discriminated between the *S.* Enteritidis and *S.* Typhimurium lineages associated with extraintestinal infections in sub-Saharan Africa[11,47]. The distribution of invasiveness index values for each lineage were compared using the Wilcoxon–Mann–Whitney test[46] implemented through R[75] v3.4.0. A custom-made database of the top 196 invasiveness predictor genes was then created from the multi-fasta file provided[47]. SRST2[76] v0.2.0 was used to flag any genes with mutations in ST313 L3 compared with ST313 L2. Exact mutations were investigated manually using MegaX[77] v0.1.

**AMR testing and statistical analysis.** Genetic determinants for AMR were identified using staramr v0.5.1 (https://github.com/phac-nml/staramr) against the ResFinder[78] and PointFinder[79] databases. Phenotypic antimicrobial susceptibility testing was performed using the EUCAST disk diffusion method[80] (Antibiotic discs: ampicillin 10 μg, chloramphenicol 30 μg and trimethoprim/sulfamethoxazole 25 μg from Mast Group). To compare the results of genotypic against phenotypic testing, sensitivity and specificity were calculated for first-line agents used over the study period using MedCalc's test evaluation calculator with Clopper–Pearson confidence intervals (https://www.medcalc.org/calc/diagnostic_test.php).

**Genomic conservation of plasmids, prophages and pseudogenes.** A literature search was used to identify genomic regions known to vary between ST19 and ST313, including plasmids[5], prophages[33] and pseudogenes[7,13–17,81,82]. We compiled a list of previously identified pseudogenes associated with the emergence of ST313 lineages in Africa (Supplementary Table 4)[14]. The collection included metabolic genes (*melR*[17] and *ttdA*[14]), environmentally responsive genes (*bcsG*[15], *katE*[15] and *macB*[13]), genes that modulate the bacterial cell surface (*flhA*[17] and *lpxO*[83]) and virulence genes (*pipD*[43], *ratB*[14], *sseI*[16]). Genes of unknown function were not included. We detected the SNPs responsible for pseudogene formation, as well as other non-synonymous mutations. A custom-made database of the identified regions was created, along with known variants. Sequences were downloaded using the online tool SalComD23580[17]. SRST2[76] v0.2.0 was then used to identify alleles present and the non-exact matches were investigated manually using MegaX[77] v0.1. Prophage and plasmid presence or absence were manually mapped onto the phylogeny using iTOL[69] v4.2. The conservation of pseudogene-associated mutations was identified at the population level and integrated into the temporal phylogenetic reconstruction using iTOL[69] v4.2. These analyses allowed inference of branches in which functional gene loss and plasmid and prophage acquisition occurred. Phenotypic testing was conducted to determine the relevance of plasmid and functional gene loss as detailed in Supplementary Methods 1.

**Phylogenetic reconstruction of pSLT plasmid.** Plasmid sequences were extracted from high-quality pseudogenomes and snp-sites[71] was used to extract SNPs. The resultant multiple sequence alignment of length 1,034 sites was used to infer a maximum-likelihood phylogeny using RAxML-ng[67] v0.6.0 with 100 bootstrap replicates to assess support. The phylogeny was visualized in iTOL[69] and convenience rooted to display lineages.

**Long-read sequencing of ST313 L3 reference strain BKQZM9.** ST313 L3 strain BKQZM9 was selected for long-read sequencing as it has the highest overall quality statistics based on short-read sequencing. An Oxford Nanopore MinION[84] 9.4.1 flowcell and SQK-RAD004 rapid sequencing kit was used with base calling by Guppy v3.1.5 (https://nanoporetech.com/nanopore-sequencing-data-analysis#). Approximately 120× coverage (600 Mb, 60,164 reads, read N50 33 kb) was generated. Hybrid genome assembly with the Illumina reads was done using Unicycler[56] v0.4.4 which resulted in a 4.8-Mb circular polished chromosome (GenBank accession: CP060169), a 94-kb circular plasmid (pSLT) (accession CP060170) and a 1,975-bp circular plasmid (pBT3) (accession: CP060171). Prokka[59] v1.12 was used for annotation as above.

**Genomic comparison of ST313 lineages.** To determine the basic architecture of the ST313 L3 BKQZM9 genome, the identity of the chromosomal and plasmid regions were compared with existing sequences from ST313 L1 (A130), ST313 L2 (D23580) and ST313 UK (U2) using the Basic Local Alignment Search Tool (BLAST)[85] and visualized using the Artemis Comparison Tool[86] v13.0.0. Putative prophage and plasmid sequences were extracted and identified using a BLAST[85] search against known ST313 prophages and plasmids and the non-redundant NCBI nucleotide database. Prophage and plasmid presence or absence were also determined experimentally (Supplementary Discussion 2).

**Reporting Summary.** Further information on research design is available in the Nature Research Reporting Summary linked to this article.

## Data availability

Sequence data that support the findings of this study have been deposited in the Sequence Read Archive (https://www.ncbi.nlm.nih.gov/sra). Accession

numbers are available in Supplementary Table 2. The complete genome and plasmid sequence for ST313 L3 strain BKQZM9 can be found under bioproject ID PRJNA656707, specifically, complete chromosome (GenBank accession: CP060169), pSLT (GenBank Accession: CP060170) and pBT3 (GenBank Accession: CP060171). The following three *S. enterica* serovar Typhimurium strains are in the process of being deposited in the DSMZ strain collection: A130 (ST313 L1), D23580 (ST313 L2) and BKQZM9 (ST313 L3). Publicly available sequence data were downloaded from one of the following sources: GenBank (https://www.ncbi.nlm.nih.gov/genbank/), Sequence Read Archive (https://www.ncbi.nlm.nih.gov/sra), European Nucleotide Archive (https://www.ebi.ac.uk/ena) or Enterobase (https://enterobase.warwick.ac.uk). Accession numbers are listed in Supplementary Table 3. The data summarized in Extended Data Fig. 2 are available in Supplementary Table 3. The remaining relevant data are within the manuscript and its supporting information files.

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

## Acknowledgements

We thank present and former members of Lab H at the Institute of Infection, Veterinary and Ecological Sciences, University of Liverpool, including members of the Hinton and Baker groups for invaluable discussions, in particular C. Chong, M. Horsburgh, S. Owen and K. Costigan for their support, contributions and advice; the MLW clinical research programme and the Institute Pasteur for access to their *Salmonella* archives; P. Ashton for helpful comments on the manuscript; and R. Low for his assistance in uploading metadata to repositories. C.V.P. is supported by the John Lennon Memorial Scholarship from the University of Liverpool and a Fee Bursary Award from the Institute of Integrative Biology at the University of Liverpool. B.K. was funded by an AESA-RISE fellowship from the African Academy of Sciences. N.H. is funded by the BBSRC, Core Strategic Programme Grant at the Earlham Institute (BB/CSP17270/1). F.-X.W. and A.T.-D. are supported by the Institut Pasteur, Santé publique and the French government's Investissement d'Avenir programme, Laboratoire d'Excellence 'Integrative Biology of Emerging Infectious Diseases' (ANR-10-LABX-62-IBEID). K.S.B. is funded by a Wellcome Trust Clinical Research Career Development Fellowship (106690/A/14/Z). J.C.D.H. is funded by a Wellcome Trust Senior Investigator Award (106914/Z/15/Z). Genome sequencing was done by the Earlham Institute as part of the 10,000 *Salmonella* genomes project which is supported by the Global Challenges Research Fund data and resources grant (BBS/OS/GC/000009D). Next-generation sequencing and library construction were delivered via the BBSRC National Capability in Genomics and Single Cell (BB/CCG1720/1) at Earlham Institute, by members of the Genomics Pipelines Group.

## Author contributions

C.V.P. was involved in conceptualizing and designing the overall study and performed the majority of phylogenetic, genomic and statistical analyses. B.M.P.-S. was involved in conceptualizing, designing and performing phenotypic characterization of isolates. R.C. contributed to conceptualization of pseudogene analysis and constructed *S.* Typhimurium 4/74 $\Delta lpxO::aph$ mutant for mass spectrometry analysis by λ red recombineering. R.P., H.C.B., L.A., M.P.M. and A.B.-B. assisted in collection of isolates from MLW Clinical Research Center. A.T.-D. assisted in the collection of isolates from Institut Pasteur. R.J. Bengtsson and W.M.P.R. contributed phylogenetic and genomic analysis and supported troubleshooting. R.J. Bennett and G.E.S. contributed to BEAST analysis and bioinformatic troubleshooting. A.Y.B., A.V.P. and A.C.D. completed long-read sequencing and assembly of ST313 L3 reference strain BKQZM9. B.K. contributed to analysis of contemporary whole-genome sequences. J.A. Bevington performed phenotypic characterization of AMR profiles. N.W. performed phenotypic characterization of the pAnkS encoded LsoA–LsoB toxin–antitoxin proteins. J.A. Bengoechea and A.D. did lipid A analysis by mass spectrometry. E.V.R. performed phenotypic confirmation of plasmid and prophage repertoire. H.J.W. and X.Z. performed phenotypic characterization of isolates. P.M.D.S. provided support in reviewing and interpreting results. N.H. sequenced isolates as part of the 10,000 *Salmonella* genomes project. F.-X.W., M.A.G., N.A.F., K.S.B. and J.C.D.H. were involved in conceptualizing and designing the overall study. K.S.B. and J.C.D.H. additionally supervised the entire study.

## Competing interests

R.C. was employed by the University of Liverpool at the time of the study and is now an employee of the GSK group of companies. The remaining authors declare no competing interests.

## Additional information

**Extended data** is available for this paper at https://doi.org/10.1038/s41564-020-00836-1.

**Correspondence and requests for materials** should be addressed to J.C.D.H.

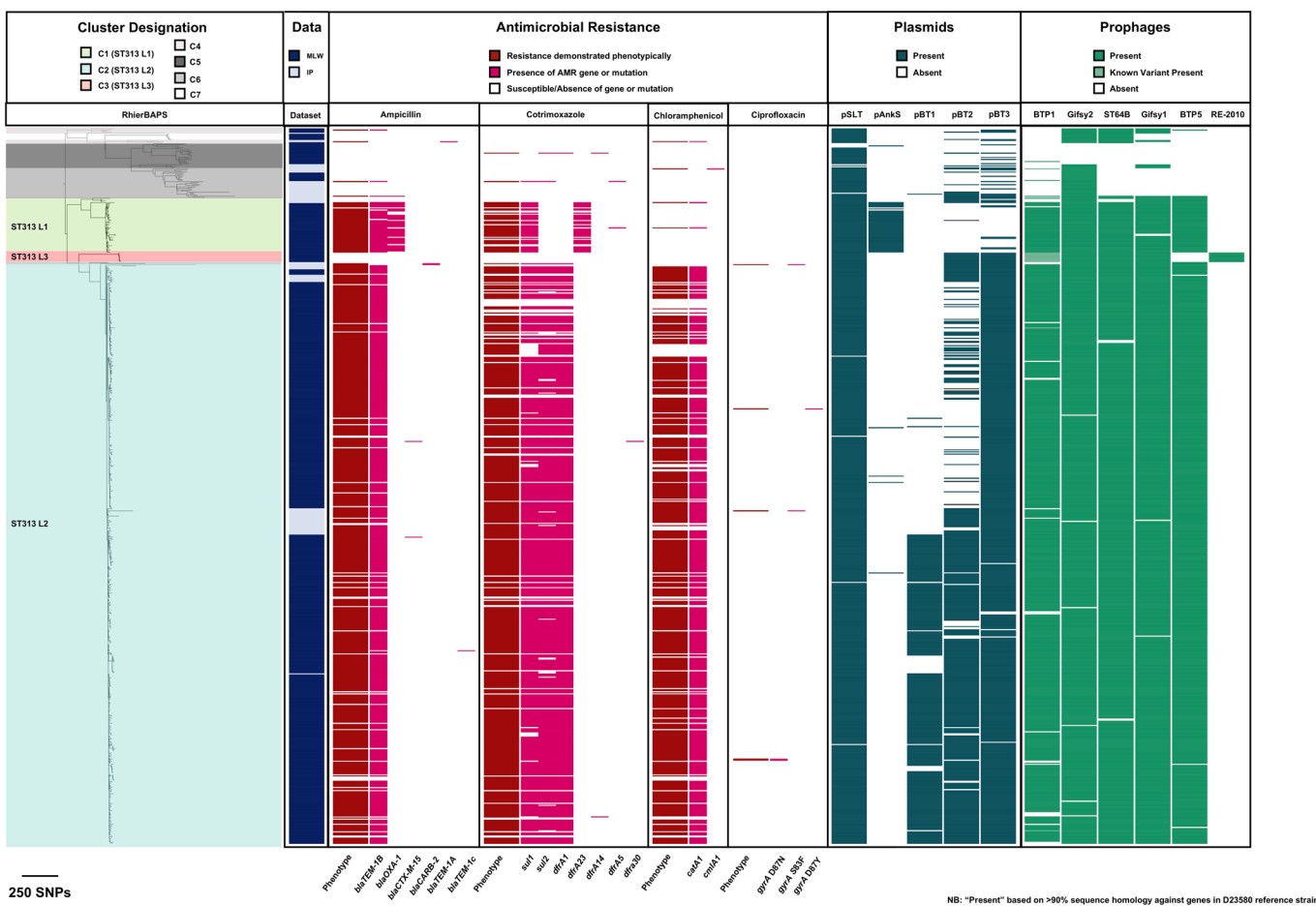

**Extended Data Fig. 1 | Dataset maximum likelihood core genome SNP phylogeny.** Maximum likelihood phylogeny based on the core genome SNP alignment of strains used in this study. Background shading on phylogeny represents cluster designation (rHierBAPs) and additional metadata is represented as adjacent colour strips. Note that Malawi-Liverpool Wellcome Clinical Research Center is abbreviated to MLW and Institute Pasteur Unité des Bactéries Pathogènes Entériques Contextual Collection is abbreviated to IP. Figure visualised using iTOL[69].

Extended Data Fig. 2 | See next page for caption.

**Extended Data Fig. 2 | Contextual maximum likelihood core gene SNP phylogeny.** Maximum likelihood phylogeny based on the core gene SNP alignment of strains in this study in the context of published ST313 genomes[5–8,11,21,22,89]. Background shading on phylogeny represents cluster designation (rHierBAPs). Outer ring provides details on the original publication. Grey represents this publication. Figure visualised using iTOL[69].

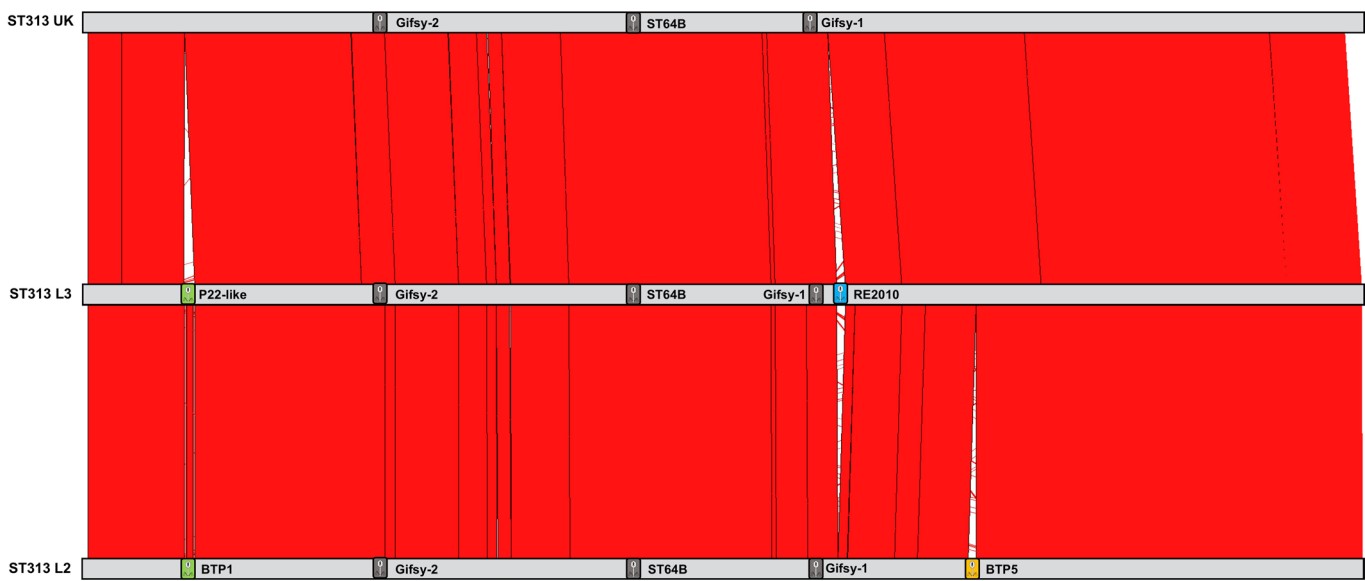

**Extended Data Fig. 3 | Chromosomal comparison of ST313 lineages.** Comparison of the complete genome sequences of ST313 lineage representatives generated using the Artemis Comparison Tool[86]. Reference isolates used; D23580 (ST313 L2), BKQZM9 (ST313 L3) and U2 (UK strains). The number of genes shared between D23580 and BKQZM9 was 4,529. An additional 167 genes were BKQZM9-specific, whereas 309 genes were exclusive to D23580, largely explained by differences in the prophage and plasmid repertoire of the two strains. Red represents sequence similarity and white represents regions absent. Prophage positions are represented as coloured boxes.

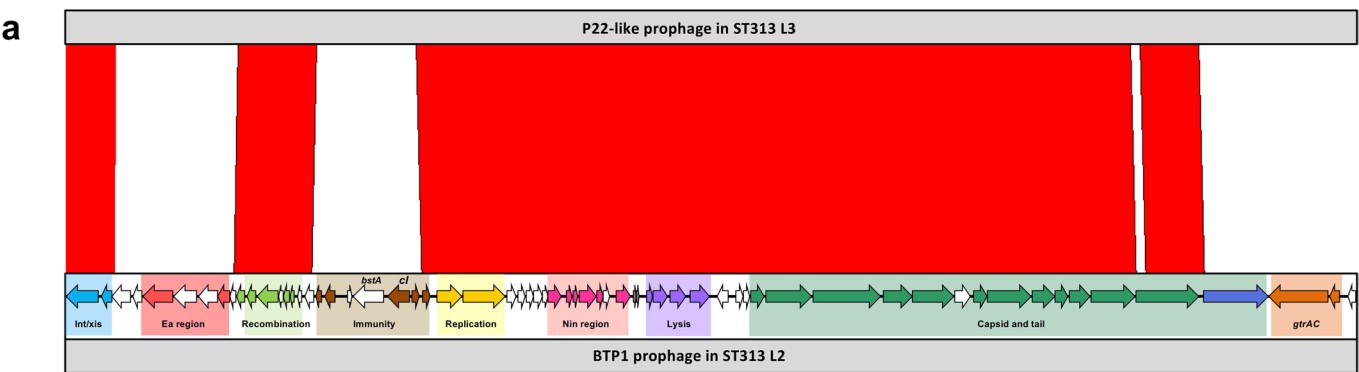

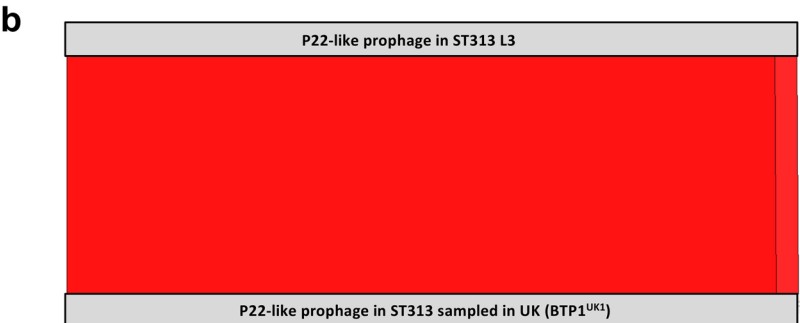

**Extended Data Fig. 4 | Prophage BTP1 comparisons.** Comparison of P22-like prophage regions identified in ST313 lineages generated using the Artemis Comparison Tool[86]. Red represents sequence similarity and white represents regions absent. **(a)** shows the comparison between P22-like prophage in ST313 L3 (strain BKQZM9) and ST313 L2 (strain D23580). Prophage annotation is adapted from that shown in Owen et al., 2017[33], with different colours highlighting different prophage regions. **(b)** shows the comparison between P22-like prophage in ST313 L3 (strain BKQZM9) and a P22-like prophage in ST313 sampled in the UK.

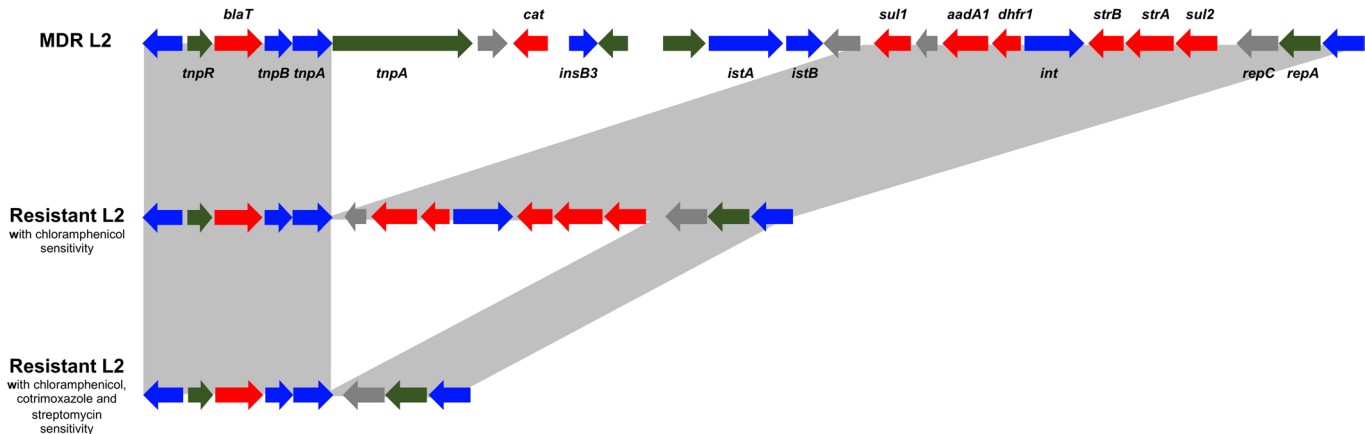

**Extended Data Fig. 5 | Examples of variation within the Tn*21*-like element.** Annotation of the resistance cassette carried on the Tn*21*-like element integrated into the *Salmonella* virulence plasmid pSLT-BT. Examples of variation in the Tn*21*-like element is shown for three different resistance profiles in ST313 L2, Grey boxes between annotations represents gene presence. Annotation is adapted from Kingsley et al., 2009[5], and shows antibiotic resistance genes (red), integrase or transposase (blue), pseudogenes (green) and other genes (grey).

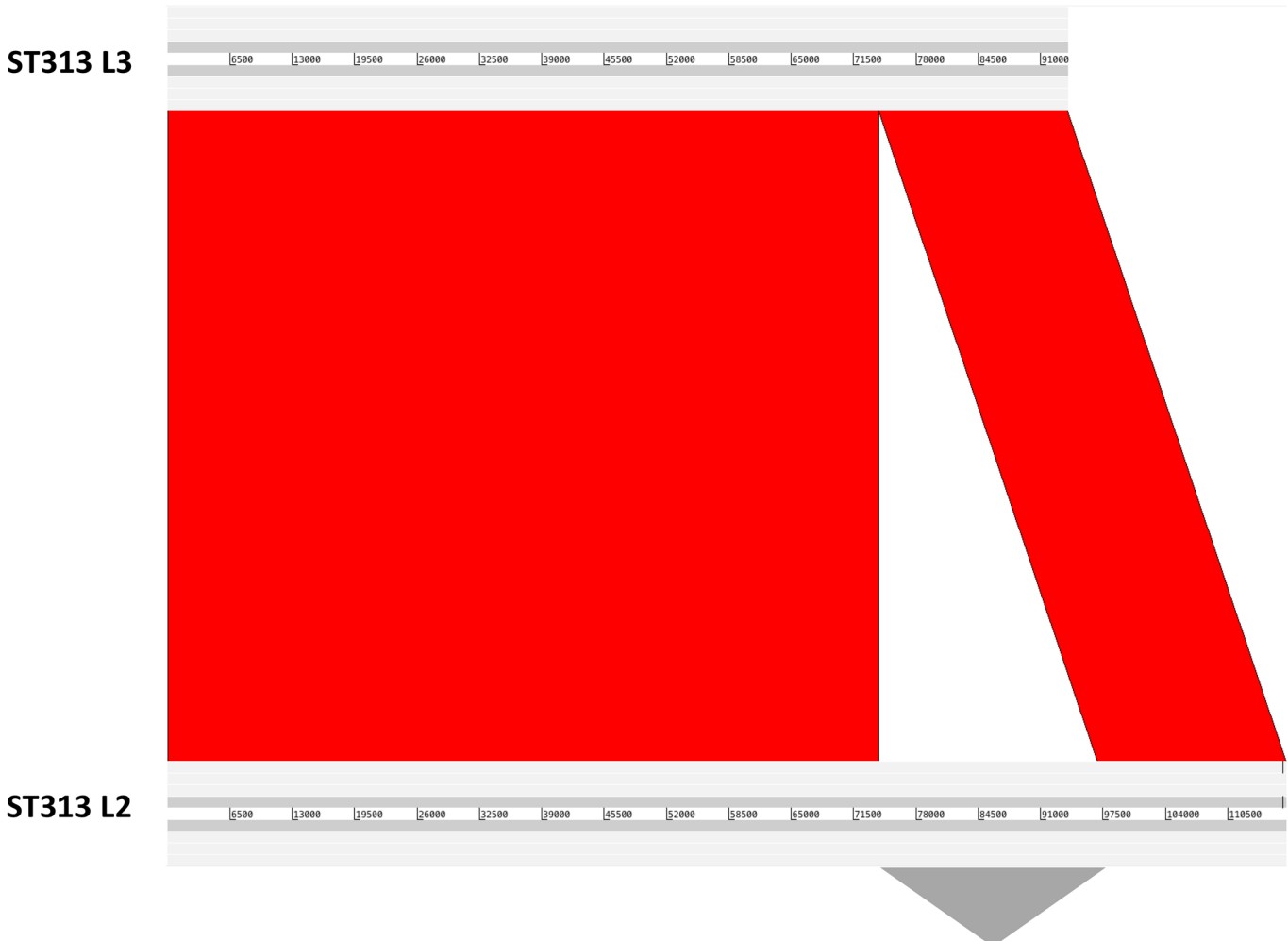

**Extended Data Fig. 6 | pSLT comparison of D23580 vs BKQZM9.** Comparison of the *Salmonella* virulence plasmid pSLT identified in ST313 L3 (BKQZM9) and ST313 L2 (D23580) generated using the Artemis Comparison Tool[86]. Red represents sequence similarity and white represents absent regions. The location of the Tn*21*-like element is indicated.

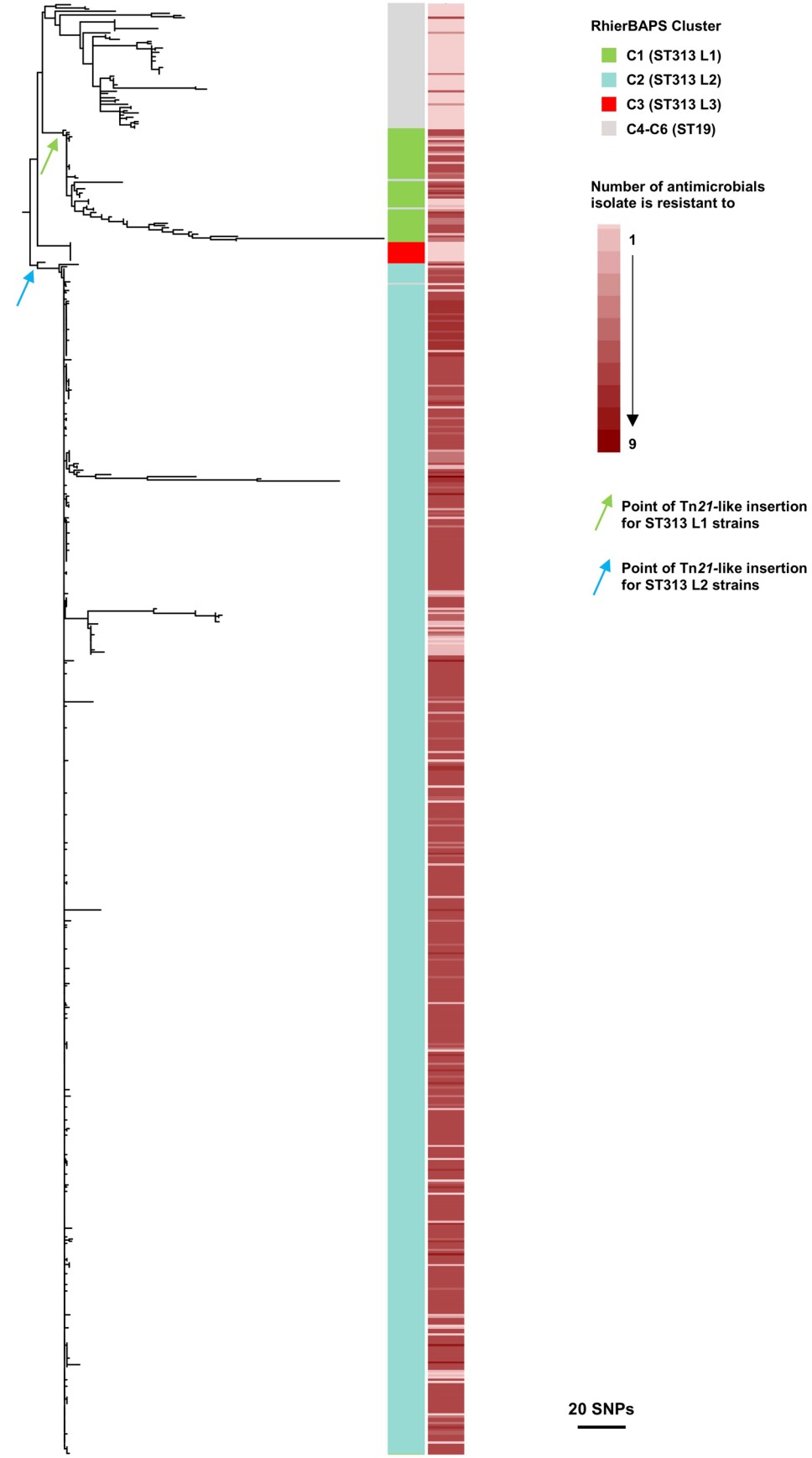

**RhierBAPS Cluster**
- C1 (ST313 L1)
- C2 (ST313 L2)
- C3 (ST313 L3)
- C4–C6 (ST19)

**Number of antimicrobials isolate is resistant to**

Point of Tn*21*-like insertion for ST313 L1 strains

Point of Tn*21*-like insertion for ST313 L2 strains

20 SNPs

**Extended Data Fig. 7 | See next page for caption.**

**Extended Data Fig. 7 | Phylogenetic reconstruction of pSLT plasmid.** Maximum likelihood phylogenetic tree showing the population structure of the pSLT plasmid in ST19 and ST313 lineages. Colour strip from right to left; cluster assignment of each isolate and number of antimicrobials each isolate is resistant to. The likely point of insertion of the Tn*21*-like element is indicated for ST313 L1 strains (green arrow) and ST313 L2 strains (blue arrow).

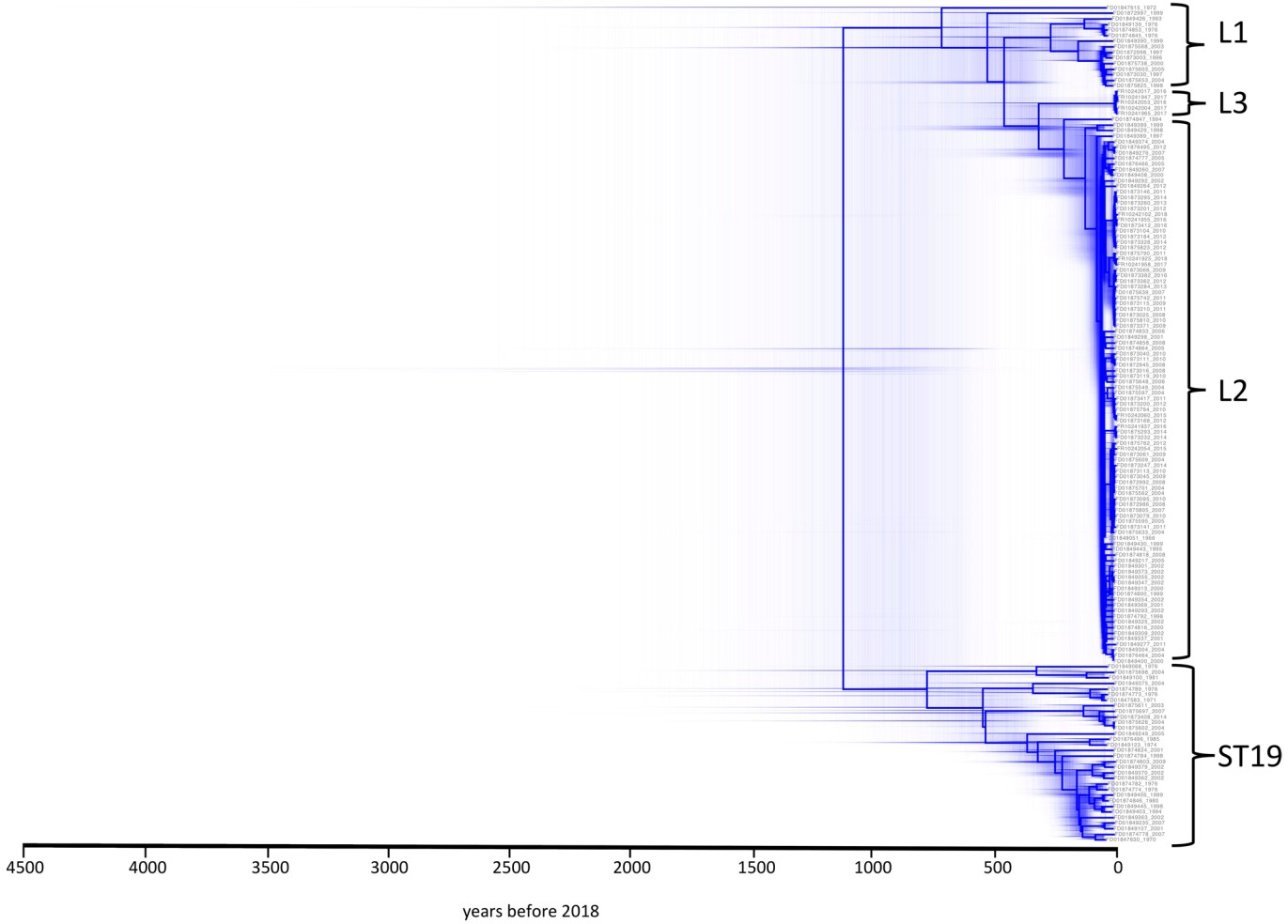

**Extended Data Fig. 8 | Distribution of possible hierarchies generated in BEAST analysis.** Chronograph of 150 *S.* Typhimurium strains isolated from bloodstream of human iNTS disease patients. The figure displays the distribution of possible hierarchies and highlights uncertainty in the timeline. Figure was visualized using DensiTree[74].

# nature research

# Reporting Summary

Nature Research wishes to improve the reproducibility of the work that we publish. This form provides structure for consistency and transparency in reporting. For further information on Nature Research policies, see our Editorial Policies and the Editorial Policy Checklist.

## Statistics

For all statistical analyses, confirm that the following items are present in the figure legend, table legend, main text, or Methods section.

| n/a | Confirmed | |
|---|---|---|
| ☐ | ☒ | The exact sample size (*n*) for each experimental group/condition, given as a discrete number and unit of measurement |
| ☒ | ☐ | A statement on whether measurements were taken from distinct samples or whether the same sample was measured repeatedly |
| ☐ | ☒ | The statistical test(s) used AND whether they are one- or two-sided *Only common tests should be described solely by name; describe more complex techniques in the Methods section.* |
| ☐ | ☒ | A description of all covariates tested |
| ☒ | ☐ | A description of any assumptions or corrections, such as tests of normality and adjustment for multiple comparisons |
| ☐ | ☒ | A full description of the statistical parameters including central tendency (e.g. means) or other basic estimates (e.g. regression coefficient) AND variation (e.g. standard deviation) or associated estimates of uncertainty (e.g. confidence intervals) |
| ☐ | ☒ | For null hypothesis testing, the test statistic (e.g. *F*, *t*, *r*) with confidence intervals, effect sizes, degrees of freedom and *P* value noted *Give P values as exact values whenever suitable.* |
| ☐ | ☒ | For Bayesian analysis, information on the choice of priors and Markov chain Monte Carlo settings |
| ☒ | ☐ | For hierarchical and complex designs, identification of the appropriate level for tests and full reporting of outcomes |
| ☒ | ☐ | Estimates of effect sizes (e.g. Cohen's *d*, Pearson's *r*), indicating how they were calculated |

*Our web collection on statistics for biologists contains articles on many of the points above.*

## Software and code

Policy information about availability of computer code

| Data collection | Software used for data collection; Microsoft Excel for Mac  v 16.29.1 |
|---|---|
| Data analysis | Software used for data analysis is as follows; Trimmomatic v0.36, Seqtk v1.2-r94, Fastqc v0.11.5, Multiqc v1.0,  Unicycler v0.3.0 and v0.4.0, QUAST v4.6.3, Prokka v1.12, SISTR v1.0.2, MLST v2.10 , BWA mem v0.7.10-r789, SAMtools v1.7, GATK v3.7, Picard v2.10.1-SNAPSHOT, Bcftools v1.9-80,  QualiMap v2.0,  Gubbins v2.2, RAxML-ng v0.6.0,  rhierBAPs v1.1.3, Interactive Tree Of Life v4.2, Roary v3.11.0,  SNP-sites v2.3.3, BEAUTI v2.6.1, BEAST2 v2.6.1, Tracer v1.7.1, LogCombiner v2.6.1, DensiTree v2.2.7, TreeAnnotator v2.6.0 , R  v3.4.0, SRST2 v0.2.0, MegaX v0.1, staramr v0.5.1, BLAST v2.6.0, Guppy v3.1.5, Artemis Comparison Tool v13.0.0, Invasiveness Index [available at https://github.com/Gardner-BinfLab/invasive_salmonella], Microsoft Excel for Mac  v 16.29.1. |

For manuscripts utilizing custom algorithms or software that are central to the research but not yet described in published literature, software must be made available to editors and reviewers. We strongly encourage code deposition in a community repository (e.g. GitHub). See the Nature Research guidelines for submitting code & software for further information.

## Data

Policy information about availability of data

All manuscripts must include a data availability statement. This statement should provide the following information, where applicable:

- Accession codes, unique identifiers, or web links for publicly available datasets
- A list of figures that have associated raw data
- A description of any restrictions on data availability

Sequence data that support the findings of this study have been deposited in the Sequence Read Archive (https://www.ncbi.nlm.nih.gov/sra). Accession numbers are available in Table S2. The complete genome and plasmid sequence for ST313 L3 strain BKQZM9 can be found under bioproject ID PRJNA656707, specifically complete chromosome (GenBank accession: CP060169), pSLT (GenBank Accession: CP060170), pBT3 (GenBank Accession: CP060171). Publicly available sequence

# Field-specific reporting

Please select the one below that is the best fit for your research. If you are not sure, read the appropriate sections before making your selection.

☒ Life sciences ☐ Behavioural & social sciences ☐ Ecological, evolutionary & environmental sciences

For a reference copy of the document with all sections, see nature.com/documents/nr-reporting-summary-flat.pdf

# Life sciences study design

All studies must disclose on these points even when the disclosure is negative.

| Sample size | Samples were derived from two archived blood culture collections to achieve the largest collection of African Salmonella Typhimurium bloodstream isolates sequenced to date.<br><br>The main dataset was sourced from from the Malawi Liverpool Wellcome Clinical Research Programme (MLW) in Blantyre, Malawi which consists of ~8,000 Salmonella Typhimurium bloodstream isolates from patients with iNTS disease. Due to budgetary and staffing limitations, the maximum number of samples which could be whole genome sequenced was 1,000.<br><br>The second collection comprised of a contextual dataset sourced from the Institut Pasteur, Paris and consists of 94 Salmonella Typhimurium isolates collected from human extraintestinal sites of patients. All 94 samples were sent for whole genome sequencing. |
|---|---|
| Data exclusions | Data was excluded from downstream analysis in the following cases;<br><br>Any bacterial sample which was unable to be resuscitated from freezer archives.<br><br>Any sample which was identified as a serovar other than Salmonella Typhimurium following computational typing of genome sequence data.<br><br>Any sample which failed quality control of sequence reads. Specifically if the sample failed basic quality statistics, per base sequence quality, per base N content, adapter content or had an average GC content out of acceptable range (47 % - 57 %).<br><br>Any assembly which failed quality control. Specifically, if the sample had an N50 > 20 kb, had 600 or more contiguous sequences or the total number of bases was outside of acceptable range (between 4Mbp and 5.8Mbp).<br><br>Any sequenced sample which failed mapping quality control. Specifically, if the mapped sample had a coverage depth less than 10x. |
| Replication | To verify study reproducibility, the results of all automated computational approaches were confirmed either at the phenotypic level and/or by using multiple bioinformatic strategies. In addition, all phenotypic testing was performed in triplicate and each of the phenotypes was fully reproducible. To generate robust maximum likelihood phylogenetic trees, bootstrapping was used to assess branch support. Bayesian evolutionary analysis (BEAST) was run on three independent chains, each of length 250,000,000. All attempts at replication were successful. |
| Randomization | For the MLW collection, isolates listed as S. Typhimurium were extracted from the metadata file and stratified by antimicrobial resistance profile into the following four categories: susceptible, resistant to one first line agent, resistant to two first line agents or multi-drug resistant. Sub-sampling was then performed on each strata using a random number generator (Excel, Microsoft) to select 1,000 isolates. Randomization was not necessary for the contextual collection, as all samples were included in the study. |
| Blinding | Data collection was performed by a diagnostic laboratory which provided extensive metadata for each isolate to facilitate contextual interpretation. Consequently, blinding was not appropriate. |

# Reporting for specific materials, systems and methods

We require information from authors about some types of materials, experimental systems and methods used in many studies. Here, indicate whether each material, system or method listed is relevant to your study. If you are not sure if a list item applies to your research, read the appropriate section before selecting a response.

## Materials & experimental systems

| n/a | Involved in the study |
|-----|----------------------|
| ☒ | Antibodies |
| ☒ | Eukaryotic cell lines |
| ☒ | Palaeontology and archaeology |
| ☒ | Animals and other organisms |
| ☒ | Human research participants |
| ☒ | Clinical data |
| ☒ | Dual use research of concern |

## Methods

| n/a | Involved in the study |
|-----|----------------------|
| ☒ | ChIP-seq |
| ☒ | Flow cytometry |
| ☒ | MRI-based neuroimaging |

