## [Peer Review File · Nature Microbiology]

Peer Review Information

Journal: Nature Microbiology

Manuscript Title: Stepwise evolution of *Salmonella* Typhimurium ST313 causing bloodstream infection in Africa.

Corresponding author name(s): Jay Hinton

Reviewer Comments & Decisions:

Decision Letter, initial version:
--

Dear Jay,

Thank you for your patience while your manuscript "The stepwise evolution of *Salmonella* Typhimurium ST313 responsible for bloodstream infection in Africa." was under peer-review at Nature Microbiology. It has now been seen by 3 referees, whose expertise and comments you will find at the end of this email. Although they find your work of some potential interest, they have raised a number of concerns that will need to be addressed before we can consider publication of the work in Nature Microbiology.

In particular, referee #1 is concerned that the "broader impact may be limited given the lack of phenotypes that provide direct and unambiguous evidence for the proposed extra-intestinal adaptation of the L3 ST313 cluster." Referee #1 suggests to add cell culture or animal experiments to extend the data on invasiveness. Referee #2 asks to include a discussion of the limitations of the bioinformatic approaches and, similar to referee #1, to validate the invasiveness index by wet lab experiments "to support that the L3 lineage does in fact disseminate more rapidly from the intestine". Referee #3 asks for several points to be clarified and for more details on the epidemiology. Referee #3 also mentions that experimental back up for the invasiveness index would strengthen the paper. Editorially, we are very interested in your study, but we feel that it will be important to address the referee concerns, especially adding experimental data to validate the invasiveness index, as requested by all three referees, and providing more details on the epidemiology of the Malawian L3 lineage.

Should further experimental data allow you to address these criticisms, we would be happy to look at a revised manuscript.

Please include a data availability statement as a separate section after Methods but before references, under the heading "Data Availability". This section should inform readers about the availability of the data used to support the conclusions of your study. This information includes accession codes to public repositories (data banks for protein, DNA or RNA sequences, microarray, proteomics data etc...), references to source data published alongside the paper, unique identifiers such as URLs to data repository entries, or data set DOIs, and any other statement about data availability. At a minimum, you should include the following statement: "The data that support the findings of this study are available from the corresponding author upon request", mentioning any restrictions on availability. If DOIs are provided, we also strongly encourage including these in the Reference list (authors, title, publisher (repository name), identifier, year). For more guidance on how to write this section please see:

<http://www.nature.com/authors/policies/data/data-availability-statements-data-citations.pdf>

- * Include a "Response to referees" document detailing, point-by-point, how you addressed each referee comment. If no action was taken to address a point, you must provide a compelling argument. This response will be sent back to the referees along with the revised manuscript.
- * If you have not done so already we suggest that you begin to revise your manuscript so that it conforms to our Article format instructions at <http://www.nature.com/nmicrobiol/info/final-submission>. Refer also to any guidelines provided in this letter.
- * Include a revised version of any required reporting checklist. It will be available to referees (and, potentially, statisticians) to aid in their evaluation if the manuscript goes back for peer review. A revised checklist is essential for re-review of the paper.

{REDACTED}

Note: This url links to your confidential homepage and associated information about manuscripts you may have submitted or be reviewing for us. If you wish to forward this e-mail to co-authors, please delete this link to your homepage first.

Nature Microbiology is committed to improving transparency in authorship. As part of our efforts in this direction, we are now requesting that all authors identified as 'corresponding author' on published papers create and link their Open Researcher and Contributor Identifier (ORCID) with their account on the Manuscript Tracking System (MTS), prior to acceptance. This applies to primary research papers only. ORCID helps the scientific community achieve unambiguous attribution of all scholarly contributions. You can create and link your ORCID from the home page of the MTS by clicking on 'Modify my Springer Nature account'. For more information please visit www.springernature.com/orcid.

If you wish to submit a suitably revised manuscript we would hope to receive it within 6 months. If you cannot send it within this time, please let us know. We will be happy to consider your revision, even if a similar study has been accepted for publication at Nature Microbiology or published elsewhere (up to a maximum of 6 months).

Reviewer Expertise:

Referee #1: Salmonella evolution, virulence, innate immunity, genomics

Referee #2: Salmonella bloodstream infections

Referee #3: Salmonella phylogenomics, genomics of pathogenesis

Reviewer Comments:

Reviewer #1 (Remarks to the Author):

In this submission, the authors perform an interesting evolutionary analysis of iNTS isolates in Malawi and the UK. Examination of a larger, contemporary collection of ST313 isolates reveals the new lineage L3, which the authors predict to exhibit signatures of adaptation to an extra-intestinal lifestyle as approximated by an elevated invasiveness index. Phylogenetic prediction suggests that L3 is genotypically and phenotypically an intermediate of the L1 and L2 ST313 lineages, and also appears to cluster with lineages from the UK. Because of the inclusion of several hundred new ST313 isolates in

the MLW collection, the authors are able to update previously reported estimates of lineage divergence dates within ST313 as well as approximate a timeline for pseudogenization across lineages.

The novelty of this work rests in the new identification of the L3 cluster of ST313 strains, as well as the authors' updated understanding of evolution across ST313. As is, this paper will be of interest to the Salmonella research community, but its broader impact may be limited given the lack of phenotypes that provide direct and unambiguous evidence for the proposed extra-intestinal adaptation of the L3 ST313 cluster. The following comments should be considered:

1. The paper could benefit from more clarification regarding the invasiveness index and its implications. Although this was previously reported in Wheeler et al 2018, the degree of predictiveness of this approach should be explained in greater detail here. From my understanding, the estimate of the invasiveness index appears to rely more heavily on loss-of-function SNPs in genes that are normally important for an intestinal lifestyle. It would be interesting if the authors discussed whether any changes within ST313 L3 suggest adaptation towards a systemic lifestyle beyond loss-of-function SNPs – i.e. beneficial changes that confer an advantage at systemic sites.

2. Fig 3 is difficult to interpret. I think that each "column" between the bubble plot and matrix represents a different AMR profile, so each segment within the bubble plot indicates the relevant abundance within each lineage per year per AMR profile? As is, the combination matrix almost looks like it is also tracking AMR through time given the order of the antibiotics/the connected lines. My recommendations for making this more clear would be to remove the connecting lines between the AMR profiles (currently they look a bit like separate lineages), replacing the "fully susceptible" line with the same dark/light grey dots bubbled in (and have all 11 dark grey), and labeling under the combination matrix plot that each column is a unique AMR signature – maybe even underneath each one you could indicate what the total # of isolates is in each AMR bin?

3. In Fig 5 it would be interesting to include similar invasiveness index prediction data for the UK lineages. It is also worth considering extending these data into cell culture or animal experiments for perhaps a representative isolate of each lineage in order to benchmark the predictive ability of the invasiveness index.

4. It would be worthwhile to include a more detailed in-text explanation of the phenotypic effects of lineage-specific SNPs to the discussion in lines 277-287 (currently only in Table 1/Table S4), and perhaps further speculation on the link between these SNPs and extraintestinal potential of L3.

Minor Comments

-Table S4 – what is the blue/pink shading indicate in the ST19/L1/L2/L3 columns in this table?

-Fig S1 – include a description of the MLW/IP abbreviations in the legend (may have missed it but no reference to IP in text? Is this the contextual dataset?) also fix extra space in "Susceptible/ Absence of gene or mutation"

-Fig S2 – "ST313 isolates in Malawi is closely related to ST313 isolated in the UK and Brazil" change is to are?

-Fig S5 – "Tn21 like element" in title add hyphen

-Line 96 remove comma in "we uncovered, a"

Reviewer #2 (Remarks to the Author):

Non-Typhi Salmonella, such as Salmonella Typhimurium and S. Enteritidis, are major causes of bloodstream infections in sub-Saharan Africa. Molecular epidemiologic studies have identified a specific multi-drug resistant sequence type (ST) of S. Typhimurium, ST313, as causing a disproportionate number of bloodstream infections compared to other sequence types. An important genomic feature of these strains is the accumulation of pseudogenes, which is a signature of alterations in the niches occupied by these strains, and this is thought to reflect human-to-human transmission in settings of poor sanitation. The manuscript by Pulford et al. advances our knowledge of the ST313 strains by analyzing genomic sequences a large set of isolates from Africa. They identify a distinct lineage of ST313, L3 that is antibiotic-susceptible and has been circulating in Malawi since 2016.

Additional analyses use bioinformatic algorithms to predict the evolution of S. Typhimurium responsible for bloodstream infections in Africa and indicating when the individual pseudogenes were acquired. Finally, the authors apply a previously reported machine learning algorithm by Wheeler and colleagues to predict the L3 lineage to be more invasive than the previously described lineages L1 and L2.

The results presented here will be of broad interest, however I think that some of the limitations of the bioinformatic approaches should be discussed (Fig. 4) and supported by additional data (Fig. 5) to increase the impact of this nice work on the field of infectious diseases.

(1) Significant strength: The authors clearly did a lot of work to analyze a huge number of genome sequences, and the implication that L3 was introduced from the UK to Africa (Fig. 2) was interesting. Figs 1-3 are major strengths and present important information on the isolates and on prevalence of antibiotic resistance that will be of broad interest.

(2) Minor weakness: There seem to be some underlying assumptions for Fig. 4 that should be discussed, such as that the mutation rate over time is constant. It seems that this might not be true, especially with the HIV epidemic in Africa that led to an increase in bacteremia cases. Would an explosion of cases not affect the mutation rate? Potential limitations of this analysis should be discussed.

(3) Weakness: The dataset analysed is biased to bloodstream infection, which would seem to limit the conclusions that can be drawn from genomic analyses with regard to relative "invasiveness" of the isolates. It was reported by Kariuki et al [PMID: 17173674] that the same strains circulating in Kenya were isolated from both, cases of bloodstream infection and gastroenteritis. If overall prevalence of a clade is higher across SSA then the number of bloodstream infections would automatically be higher. However, there seems to be relatively little banking of intestinal isolates to assess "invasiveness". Therefore, I think the premise that invasiveness can be assessed by looking at the genome of only bloodstream isolates, in the absence of data on overall infections, is flawed. The denominator of total

cases (bloodstream + intestinal infection) is missing. This should be discussed as a shortcoming of analyzing bloodstream isolates only.

(4) Weakness: the data in Fig. 5 on "invasiveness index" should be validated by wet lab experiments to support that the L3 lineage does in fact disseminate more rapidly from the intestine. If animal experiments would support this conclusion, it would significantly increase the impact of this analysis on the field.

Reviewer #3 (Remarks to the Author):

This paper describes aspects of the evolution of mainly blood stream isolates of *S. Typhimurium* with a particular focus on isolates from Malawi and Sub-Saharan Africa. Such isolates are dominated by those of a sequence type known as ST313 with a minority belonging to ST19. ST313 isolates were originally classified through phylogenomics into 2 main lineages (L1 and L2) that were associated with antibiotic resistance and a recent spread across the African continent, associated with the emergence of HIV. ST313 have particular genome structures and exhibit evidence of genome degradation that can be linked to possible host adaptation to human transmission potential. Here the authors provide further insight into L1 and L2 and also define a new Malawi-associated lineage they call L3. L3, interestingly is related to similar lineages from the UK and Brazil and is drug sensitive.

This is an interesting paper written very clearly and beautifully illustrated. There are two main contributions (a) further definition of the phylogenetic structure of the ST313 group and (b) definition of a new lineage associated with Malawi known as L3. For point (a) the main conclusions are around the ages of the lineages as assessed using BEAST-type analysis

For point (b) the main questions are.

1. The choice of comparator group is a bit odd. Malawi is in East Africa but most of the comparator isolates are from West Africa. Why was this set chosen? Why not use isolates closer to hand e.g. from Kenya etc where there are many sequenced isolates available? The actual number of contextual isolates is only 43 ST313 spread over several countries.
2. When comparing D23580 and BKQZM9 are the plasmid genes included in the comparison of 'unique genes'? It looks as if the only differences are either prophage or plasmid genes aside from the small indels and SNPs mentioned elsewhere?
3. Can the authors say anything about the geospatial distribution of the L3 lineage in Malawi? Do they have GPS data? It looks as if this is an outbreak recently introduced into the Blantyre region. How large is the geographical region covered? Could they provide a map of the outbreak?
4. It looks as if L3 has emerged in part through recombination or deletions across prophage regions and plasmid loss/exchange. Looking at the tree the Malawian isolates are on a significant branch away from the UK and Brazilian isolates. Could the authors provide more details about the differences between these three sub-clades of L3? Do they know where the indicator/first case of L3 might be in Malawi?
5. It is interesting that L3 is sensitive. Have the authors built separate phylogeny of the pSLT plasmid? How does the plasmid relate across the tree, between the lineages? Also, sometimes when a transposon is lost there is a 'scar' left at the point of deletion. Is the L3 lineage a loss of a transposon

or the introduction of an pSLT plasmid that never harboured a transposon.

Other general comments

There is a lot of work in the paper describing the antibiotic resistance including fluoroquinolone resistance yet the L3 clade is sensitive. Is all of this discussion on AMR needed? It does not look particularly novel and does not add much to the paper.

Also, the measurement of an 'invasive index' is interesting but there is no experimental back up in the context of invasion or have I missed something? As ST L3 is lacking many effectors associated with salmonella effectors a simple cellular invasion assay would be interesting?
Could the authors speculate about the biotyping results? What does this mean biologically?

Author Rebuttal to Initial comments

Reviewer 1 Comments

The paper could benefit from more clarification regarding the invasiveness index and its implications. Although this was previously reported in Wheeler et al 2018, the degree of predictiveness of this approach should be explained in greater detail here. From my understanding, the estimate of the invasiveness index appears to rely more heavily on loss-of-function SNPs in genes that are normally important for an intestinal lifestyle.

We are grateful for the suggestion, and agree that a more detailed explanation regarding the implications of the invasiveness index would be useful. We have added a more thorough rationale for using this tool (lines 371 to 382). Specifically, we have stated;

“As detailed above, genome degradation is a key feature in the evolution of bacterial pathogens that are adapted to systemic infection. By quantifying the extent of genome degradation in ST313 pathovariants we aimed to investigate the extraintestinal potential of the different lineages. Initial attempts to measure pseudogenisation have been labour intensive, relying on the manual annotation of truncations, deletions and frameshifts in genes^{61,62}. Recently, the “invasiveness index” was defined as a value that represents the extent of genome degradation and diversifying-selection specific to invasive serovars using a proven set of extraintestinal predictor genes⁶³. The machine-learning approach uses a random forest classifier and delta-bitscore functional variant-calling to discriminate between the genomes

of invasive and gastrointestinal *Salmonella*. The invasiveness index has been validated using multiple *Salmonella* serovars, and the approach clearly discriminated between the *S. Enteritidis* and *S. Typhimurium* lineages associated with extraintestinal infections in sub-Saharan Africa^{13,63}.”

It would be interesting if the authors discussed whether any changes within ST313 L3 suggest adaptation towards a systemic lifestyle beyond loss-of-function SNPs – i.e. beneficial changes that confer an advantage at systemic sites.

We agree that a search for interesting genes acquired by ST313 L3 was worthwhile. We did a detailed comparison of the gene content of ST313 L1, L2, L3 and UK-ST313 reference isolates, and identified 116 genes that were unique to ST313 L3. The vast majority of the ST313 L3-specific genes are associated with the RE-2010 prophage; all other major lineage-specific differences are examples of genome degradation which were already highlighted (lines 398 to 407). This finding reflects mounting evidence in the literature that shows the majority of adaptation of ST313 to a systemic lifestyle is driven by genome degradation. To clarify this point in the manuscript, we have added the following text (lines 458-461);

“The majority of ST313 L3-unique genes were plasmid or prophage-encoded, including the prophage RE-2010⁴² which is found in other extraintestinal *Salmonella* pathovariants⁴³⁻⁴⁵. Aside from accessory genome composition, the majority of ST313 L3-specific changes involve loss-of-function SNPs in genes that are not required for systemic infection.”

Fig 3 is difficult to interpret. I think that each “column” between the bubble plot and matrix represents a different AMR profile, so each segment within the bubble plot indicates the relevant abundance within each lineage per year per AMR profile? As is, the combination matrix almost looks like it is also tracking AMR through time given the order of the antibiotics/the connected lines. My recommendations for making this more clear would be to remove the connecting lines between the AMR profiles (currently they look a bit like separate lineages), replacing the “fully susceptible” line with the same dark/light grey dots bubbled in (and have all 11 dark grey), and labeling under the combination matrix plot that each column is a unique AMR signature – maybe even underneath each one you could indicate what the total # of isolates is in each AMR bin?

We understand concerns regarding the complexity of Figure 3, and the reviewer's recommendations prompted us to update the figure. To improve interpretation of the diagram, we replaced the "fully susceptible" line with the same dark/light grey dots (the fully-susceptible column now appears as light grey dots to indicate no resistance). The vertical connecting lines have been retained for clarity. We have added the total number of isolates to each AMR bin, and provided a label beneath the combination matrix plot stating that "Each column represents a unique AMR signature".

In Fig 5 it would be interesting to include similar invasiveness index prediction data for the UK lineages. It is also worth considering extending these data into cell culture or animal experiments for perhaps a representative isolate of each lineage in order to benchmark the predictive ability of the invasiveness index.

We are grateful for the excellent suggestion to include the UK lineages in the invasiveness index prediction. We have added this information, and updated Figure 5 accordingly. We have also added text (line 393-395) to explain that ST313 L3 had a significantly greater invasiveness index than the UK-isolated ST313 (median=0.134, SD=0.018)(W=480, p-value<0.001).

It would be worthwhile to include a more detailed in-text explanation of the phenotypic effects of lineage-specific SNPs to the discussion in lines 277-287 (currently only in Table 1/Table S4), and perhaps further speculation on the link between these SNPs and extraintestinal potential of L3.

We are grateful for the reviewers' suggestions. We have expanded the discussion (lines 333-368), including information from Table 1 and Table S4. The new section reads as follow:

"To complement our genome-derived evolutionary insights, we investigated clinical *S. Typhimurium* isolates that had genetic differences in specific genes. These experiments allowed us to link a number of lineage-specific phenotypes to individual SNPs (summarised in Table 1 and detailed fully in Table S4). For example, the KatE catalase protects bacteria from oxidative stress in the environment, and has a high level of activity in stationary phase cultures of *S. Typhimurium* ST19¹⁷. It was reported previously that the E117G KatE mutation was responsible for a low level of catalase activity in ST313 L1, ST313 L2, ST313 L2.1 and UK-isolated ST313^{9,13,17}, and we report here a similarly low activity in ST313 L3 (Table 1, Table S4). Catalase is required for multicellular growth as biofilms⁵⁴, a phenotype associated

with survival outside the host, leading to the suggestion that pseudogenisation of *katE* in ST313 lineages reflects adaptation to a more restricted host-range¹⁶.

The RDAR-negative (red, dry and rough) phenotype of ST313 L2⁵⁵ is caused by a combination of the pseudogenisation of *bcsG*¹⁷ (a gene that encodes a cellulose biosynthetic enzyme required for biofilm formation) and two nucleotide changes in the *csgD* gene promoter region⁵⁵. Here we show that ST313 L3 has an intermediary RDAR phenotype, with an incomplete wrinkling pattern on the colony surface. Although *bcsG* was not a pseudogene in ST313 L3, we identified a single SNP in the *csgD* gene promoter region which corresponded to the -189 promoter mutation described in ST313 L1 and L2 by MacKenzie *et al.* (2019)⁵⁵. The loss of traits required for stress-resistance and biofilm formation by ST313 L3 is consistent with a reduced requirement for environmental survival, and mounting evidence for human-human transmission of ST313^{56,57}.

Adaptation to an invasive lifestyle is associated with a decreased ability to colonise and survive within the gastrointestinal tract^{5,7}. *S. Typhimurium* ST19 relies upon carbon metabolism to colonise the mammalian gastrointestinal tract⁵⁸ whereas genes required for utilisation of specific carbon sources, such as tartrate and melibiose, are not functional in ST313 lineages¹⁹. We found that ST313 L3 was unable to grow with tartrate or melibiose as a sole carbon source, consistent with the pseudogenisation of the *ttdA* and *melR* genes in both ST313 L3 and ST313 L2^{5,19} (Figure 4).

Further evidence of genome degradation, consistent with niche adaptation, is provided by the pseudogenisation of the genes *ratB* in all ST313 lineages, *pipD* in ST313 L2 and L3, and *lpxO* in ST313 L2. The PipD effector protein has been implicated in gastrointestinal pathogenesis of ST19, although a causal relationship has not been demonstrated^{53,59}. The *lpxO* gene is pseudogenised by a stop codon in ST313 L2⁵. LpxO hydroxylates lipid A, a modification required for virulence of *S. Typhimurium* ST19⁶⁰. Here we used mass spectrometry to show that the lack of functional LpxO caused structural modifications of Lipid A in ST313 L2 (Table 1 and Supp Table 4). The *lpxO* gene is functional in ST313 L3, reflecting a distinct evolutionary path for this lineage.”

Table S4 – what is the blue/pink shading indicate in the ST19/L1/L2/L3 columns in this table?

We have added a footnote to Table S4 stating;

“Note that predicted functionality (see Methods) is depicted as a colour strip for each gene and is based on whole genome-based predictions of SNPs likely to play a functional role. Specifically, blue indicates functional, and pink indicates non-functional.”

Fig S1 – include a description of the MLW/IP abbreviations in the legend (may have missed it but no reference to IP in text? Is this the contextual dataset?) also fix extra space in “Susceptible/ Absence of gene or mutation”

Thank you, we have made these amendments to Figure S1.

Fig S2 – “ST313 isolates in Malawi is closely related to ST313 isolated in the UK and Brazil” change is to are?

We have changed “is to “are” in Figure S2.

Fig S5 – “Tn21 like element” in title add hyphen

We have added the hyphen to Tn21-like.

Line 96 remove comma in “we uncovered, a”

We have removed the comma.

Reviewer 2 Comments

Minor weakness: There seem to be some underlying assumptions for Fig. 4 that should be discussed, such as that the mutation rate over time is constant. It seems that this might not be true, especially with the

HIV epidemic in Africa that led to an increase in bacteremia cases. Would an explosion of cases not affect the mutation rate? Potential limitations of this analysis should be discussed.

Reviewer 2 raises an important question. It is true that the HIV epidemic in Africa may have affected the mutation rate over time, and across the phylogenetic tree. To account for rate heterogeneity amongst branches during the temporal analysis, we used a relaxed log normal clock rate which allows each branch to have its own independent rate drawn from a shared log-normal distribution. To clarify this point, and to highlight the limitations of Bayesian inference, we have included the following text (lines 275 to 277).

“We included the prior assumptions of a coalescent Bayesian skyline model for population growth, and a relaxed log normal clock rate to account for rate heterogeneity amongst branches (the full model is described in Data S1). Based on these prior assumptions...”

We have also added the following to lines 284 to 286;

“A limitation of Bayesian analysis is that it relies upon a series of user-defined prior probability distributions which influence the estimation of tip dates.”

Weakness: The dataset analysed is biased to bloodstream infection, which would seem to limit the conclusions that can be drawn from genomic analyses with regard to relative “invasiveness” of the isolates. It was reported by Kariuki et al [PMID: 17173674] that the same strains circulating in Kenya were isolated from both, cases of bloodstream infection and gastroenteritis. If overall prevalence of a clade is higher across SSA then the number of bloodstream infections would automatically be higher. However, there seems to be relatively little banking of intestinal isolates to assess “invasiveness”. Therefore, I think the premise that invasiveness can be assessed by looking at the genome of only bloodstream isolates, in the absence of data on overall infections, is flawed. The denominator of total cases (bloodstream + intestinal infection) is missing. This should be discussed as a shortcoming of analyzing bloodstream isolates only.

We agree that the measurement of invasiveness index should not be assessed by looking at the genome of only bloodstream isolates. Consequently, we ran the invasiveness index using a model that

was pre-trained by Wheeler *et al.* – a model that was based on a mixture of gastrointestinal and extraintestinal salmonellae. To highlight this point, we have added the following text (lines 386 to 388).

“Because the isolates used in this study originated from human bloodstream, we used the invasiveness index model that was pre-trained by Wheeler *et al.*, using a mixture of gastrointestinal and extraintestinal salmonellae⁵¹.”

Weakness: the data in Fig. 5 on “invasiveness index” should be validated by wet lab experiments to support that the L3 lineage does in fact disseminate more rapidly from the intestine. If animal experiments would support this conclusion, it would significantly increase the impact of this analysis on the field.

We agree with the reviewer that an animal model would be ideal to provide unambiguous evidence supporting an increased level of invasiveness in ST313 L3. However, there is currently no suitable model available for ST313 L3. We have provided a thorough discussion of the implications of this extremely important point in the text (lines 401 to 422);

“Clearly, the use of a genome-based machine learning approach does not prove that ST313 L3 has adapted to an extraintestinal lifestyle. In the past, infection models have been important for understanding the virulence and systemic spread of *Salmonella* pathovariants that cause gastroenteritis⁶⁸. However, previously cellular and animal infection models have failed to discriminate between levels of invasiveness of L1 and L2⁷. This problem was exemplified in a recent study which defined the invasiveness index of a new ST313 sublineage identified in the Democratic Republic of Congo. The study by Van Puyvelde and colleagues¹³ showed that the murine whole-animal infection model and the human macrophage cellular infection model did not find robust differences in terms of invasive disease between the different ST313 lineages and sublineages.

Several lines of evidence suggest that ST313 is in the process of becoming a human-adapted pathogen, prompting concerns about the use of animals to assess the levels of invasiveness of ST313. A similar issue arose for the invasive pathogen *Salmonella* Typhi, leading researchers to develop a human infection model that proved that the typhoid toxin is not required for

infection by *S. Typhi*⁶⁹. However, no human challenge model has yet been developed for ST313.

Although the invasiveness index cannot be experimentally validated at the moment, we note that the *Salmonella* isolates that vary in terms of invasiveness index do produce distinct clinical symptoms in a human population¹⁴. The development and validation of experimental approaches for the robust measurement of the invasiveness of *S. Typhimurium* will be an important focus for future research, but goes beyond the scope of this paper. Meanwhile, the “invasiveness index” is a useful tool for monitoring the genetic signatures associated with invasiveness and host adaptation in ST313 lineages.”

Reviewer 3 Comments

The choice of comparator group is a bit odd. Malawi is in East Africa but most of the comparator isolates are from West Africa. Why was this set chosen? Why not use isolates closer to hand e.g. from Kenya etc where there are many sequenced isolates available? The actual number of contextual isolates is only 43 ST313 spread over several countries.

The reviewer has correctly commented that we have largely focused on contextual isolates from Western Africa. There are numerous studies which have focused on understanding ST313 in Central and Eastern Africa, which have been cited in Supplementary Figure 2. The inclusion of comparator isolates from Western Africa has expanded the known geographic range of ST313 across Africa. We have added the following text (lines 138-143) to highlight this point;

“Previous genome-based analyses of the ST313 epidemic in Africa have included isolates from Eastern and Central Africa including Malawi, Kenya, Uganda, Democratic Republic of Congo and Mozambique^{6,13}. Here, by including a large range of countries, in particular from Western Africa, we have expanded the known geographic range of ST313^{6,10,13,27-30} to include Cameroon, Central African Republic, Niger, Senegal, Sudan and Togo, although there were relatively few isolates from some countries.”

When comparing D23580 and BKQZM9 are the plasmid genes included in the comparison of ‘unique

genes? It looks as if the only differences are either prophage or plasmid genes aside from the small indels and SNPs mentioned elsewhere?

Both chromosomal and plasmid genes were included in the analysis, we thank the reviewer for pointing this out and have updated the text at line 158 accordingly.

Can the authors say anything about the geospatial distribution of the L3 lineage in Malawi? Do they have GPS data? It looks as if this is an outbreak recently introduced into the Blantyre region. How large is the geographical region covered? Could they provide a map of the outbreak?

Malawi isolates were sampled from patients attending the Queen Elizabeth Central Hospital (QECH). QECH is the largest government hospital in Malawi and provides free healthcare to approximately 10,000 adults and 50,000 paediatric inpatients per year. The catchment area covers the Blantyre district and the Southern region of Malawi, but unfortunately patient-level GPS data is unavailable. To speculate on ST313 L3 being a potential outbreak, we have calculated the maximum pairwise difference between strains and added a few sentences (lines 165-168) explaining that GPS data is unavailable;

“Isolates in ST313 L3 are closely related, with a maximum pairwise distance of 16 SNPs. To determine if ST313 L3 represents an outbreak in Malawi, it would be necessary to combine this genome-based information with details on time, person and place. However, patient-level GPS data were unavailable, meaning that the geospatial distribution of ST313 L3 remains unknown.”

It looks as if L3 has emerged in part through recombination or deletions across prophage regions and plasmid loss/exchange. Looking at the tree the Malawian isolates are on a significant branch away from the UK and Brazilian isolates. Could the authors provide more details about the differences between these three sub-clades of L3? Do they know where the indicator/first case of L3 might be in Malawi?

The reviewer is correct that the main difference between ST313 L3 and UK-related ST313 is the accessory genome, specifically involving the prophage and plasmid repertoire which we have highlighted in Figure 2. The phylogenetic tree in Figure S2 was built using a core gene SNP alignment, which by definition excludes parts of the accessory genome such as prophage and plasmid regions.

As the reviewer observed, the Malawian isolates remain on a significant branch away from the UK and Brazilian isolates. These SNPs are captured in the differing invasiveness indices of the ST313 L3 and UK-ST313 strains, revealing additional loss-of function SNPs in ST313 L3 compared with UK-related ST313. We have updated Figure 5 to highlight the differences in invasiveness index between L3 and UK-related isolates.

It is interesting that L3 is sensitive. Have the authors built separate phylogeny of the pSLT plasmid? How does the plasmid relate across the tree, between the lineages? Also, sometimes when a transposon is lost there is a 'scar' left at the point of deletion. Is the L3 lineage a loss of a transposon or the introduction of an pSLT plasmid that never harboured a transposon.

This was an excellent suggestion and we have now included a phylogenetic reconstruction of the pSLT plasmid, which shows four separate clades that represent ST313 L1, L2, L3 and ST19 isolates. We additionally performed a manual investigation of the plasmid sequence, and did not identify any scars associated with transposon excision. We have provided the phylogenetic tree as a new supplementary figure (Figure S7) and added the following text to the manuscript (lines 251-256);

“Phylogenetic reconstruction of the pSLT plasmid (Figure S7) reflected the core genome SNP-based phylogeny, demonstrating four individual phylogenetic clades (ST313 L1, L2, L3 and ST19). No evidence of scars associated with plasmid excision at the usual insertion site of the Tn21-like transposable element on the pSLT plasmid were found in ST313 L3. Taken together, these findings are consistent with the pSLT plasmid of ST313 L3 never having harboured a Tn21 transposon.”

There is a lot of work in the paper describing the antibiotic resistance including fluoroquinolone resistance yet the L3 clade is sensitive. Is all of this discussion on AMR needed? It does not look particularly novel and does not add much to the paper.

We have removed the final three paragraphs of discussion of AMR, and moved it into Supplementary text 2.

Also, the measurement of an 'invasive index' is interesting but there is no experimental back up in the

context of invasion or have I missed something? As ST L3 is lacking many effectors associated with salmonella effectors a simple cellular invasion assay would be interesting?

We agree with the reviewer that an animal model would be ideal to provide unambiguous evidence supporting an increased level of invasiveness in ST313 L3. Please see response to comments from Reviewer 1.

Could the authors speculate about the biotyping results? What does this mean biologically?

We have included a more thorough discussion on the results from our phenotypic testing (lines 333-363). Please see response to comments from reviewer 1.

Decision Letter, first revision:

Dear Caisey and Jay,

Thank you for your patience while your manuscript "The stepwise evolution of *Salmonella* Typhimurium ST313 responsible for bloodstream infection in Africa." was under peer review at Nature Microbiology. It has now been seen by our referees, and in the light of their advice I am delighted to say that we can in principle offer to publish it. First, however, we would like you to revise your paper to ensure that it is in Nature Microbiology format.

Editorially, we will need you to make some changes so that the paper complies with our Guide to Authors at <http://www.nature.com/nmicrobiol/info/gta>.

Nature Microbiology offers a transparent peer review option for new original research manuscripts submitted from 1st December 2019. We encourage increased transparency in peer review by publishing the reviewer comments, author rebuttal letters and editorial decision letters if the authors agree. Such peer review material is made available as a supplementary peer review file. **Please state in the cover letter 'I wish to participate in transparent peer review' if you want to opt in, or 'I do not wish to participate in transparent peer review' if you don't.** Failure to state your preference will result in delays in accepting your manuscript for publication.

In recognition of the time and expertise our reviewers provide to Nature Microbiology's editorial process, we would like to formally acknowledge their contribution to the external peer review of your manuscript entitled "The stepwise evolution of *Salmonella* Typhimurium ST313 responsible for bloodstream infection in Africa.". For those reviewers who give their assent, we will be publishing their names alongside the published article.

I appreciate this email is long and recommend that you print it and use it as a checklist, reading it carefully to the end, in order to avoid delays to publication down the line.

Please note that we will be considering your paper for publication as an ARTICLE in our pages.

Specific points:

In particular, while checking through the manuscript and associated files, we noticed the following specific points which we will need you to address:

1. Length. At 5,852 words, your manuscript currently exceeds our normal length limit for Articles of about 3,000 words. We have some flexibility, and can allow a revised manuscript at 3,500 words, but please consider this a firm upper limit. You could achieve some shortening by moving some details to the Methods section that should follow the main text (the length of the Methods section is unlimited and does not count towards the main text length).

2. Abstract. The current abstract, at 256 words, is too long. Article abstracts should be 150-200 words. Furthermore, we suggest to edit the abstract to include the size of the strain collection, the number of genomes and the timespan of the entire collection (50 years).

3. Main text display items and supplementary information. Please note that we have recently started publishing additional figures as "Extended Data". These figures appear online in the html version of the manuscript in the place they are referred to and greatly increase discoverability of the data that is presented in them.

All Supplementary Information must be submitted in accordance with the instructions in the attached Inventory of Supporting Information, and should fit into one of three categories: EXTENDED DATA (ED); SUPPLEMENTARY INFORMATION (SI); and SOURCE DATA. Below are detailed instructions on how to format each category. For your paper, we suggest that you do the following:

a. Main figures: please maintain the current 5 main figures that illustrate the main findings of the paper.

b. Extended data (ED): please convert the 8 SI figures into ED figures. These are an integral part of the paper (presented online in the online version) and are meant to be multipanel A4 size figures. More information on file formats and how the legends should be supplied can be found below and in the attached Inventory of Supporting Information.

c. Supplementary information (SI): your study will have the 'Supplementary online data' as SI. Please submit all SI as a separate pdf file. All supplementary materials need to be assembled into a single file, including all tables (excluding those that are excessively large). In the Supplementary Information file, figure legends should be immediately below each figure and the pages should be numbered.

d. Source data: this format should be used to display source data linked to the main figures and ED figures.

We strongly encourage you include as much additional raw data underlying the graphs in the main and ED figures as possible. These data should be supplied as Excel tables, one file per main or ED figure, and should be clearly labeled and presented in a way that individual experiments are identifiable (for example, across a time course if applicable).

4. Data Availability statement. The data availability statement should clearly refer to all of the source data provided in the manuscript (more instructions on how to write this section can be found in the general formatting guidelines below). Furthermore, please add the relevant information to the 'data availability statement' and also in the table of all the strains (eg. indicating which ones will be available from a repository).

5. Reporting checklist. Please revise this document according to the instructions found in the annotated PDF attached to this message and send in a final version with your article. The final reporting checklist will be published with your manuscript.

6. Data deposition. Please carefully check through the manuscript whether all different types of sequence data have been deposited in appropriate databases.

7. Competing interest statement. The competing interest statement needs to be included in the manuscript text (before or after the Acknowledgements).

8. Author contributions. Please provide a more detailed and specific author contributions statement. A good example can be found at the end of the following article
<http://www.nature.com/nature/journal/v532/n7599/full/nature17433.html>

9. New/Novel. There are instances of the use of the word novel in the text (for example, line 370). Please remove it, except if it is strictly necessary. It is journal policy to limit unnecessarily hyperbolic use of terms related to novelty.

10. ORCID. We now require corresponding authors to provide an ORCID identifier, and would ask that you please provide one with your final submission (please also see below). There is a step during the upload of the information to our online system in which the number can be introduced.

11. Replicates and statistics. While carefully checking the figures, we noted a few things that need to be revised so that they comply with our style guidelines and accurately report on the number of replicates, statistical testing, etc. As general rules, please note that:

General comments:

Wherever statistics have been derived (e.g., error bars, box plots, statistical significance), the legend needs to provide and define the n number (i.e., the sample size used to derive statistics) as a precise value (not a range), using the wording "n=X biologically independent samples/animals/independent experiments," etc. as applicable.

All error bars need to be defined in the legends (e.g., SD, SEM) together with a measure of centre

(e.g. mean, median), and should be accompanied by their precise n number defined as noted above.

All violin plots need to be defined in the legends in terms of minima, maxima, centre, and percentiles, and should be accompanied by their precise n number defined as noted above.

The figure legends must indicate the statistical test used and if applicable, whether the test was one- or two-sided. A description of any assumptions or corrections such as tests of normality and adjustment for multiple comparisons must also be included.

For null hypothesis testing, please indicate the test statistic (e.g., F, t, r) with confidence intervals, effect sizes, degrees of freedom and P values noted.

Test results (e.g., p-values, q-values) should be given as exact values whenever possible and appropriate, and confidence intervals noted.

Please indicate how estimates of effect sizes were calculated (e.g., Cohen's d, Pearson's r).

Please state in the legends how many times each experiment was repeated independently with similar results. This is needed for all experiments but is particularly important wherever representative experiments are shown. If space in the legends is limiting, this information can be included in a section titled "Statistics and Reproducibility".

For all bar graphs, the corresponding dot plot must be overlaid.

Specific comments to address:

Please see the attached "Extended_comments" file.

General points:

Please read carefully through all of the following general formatting points when preparing the final version of your manuscript, as submitting the manuscript files in the required format will greatly speed the process to final acceptance of your work.

Titles should give an idea of the main finding of the paper and ideally not exceed 90 characters (including spaces). We discourage the use of active verbs and do not allow punctuation.

The paper's summary paragraph (about 150-200 words; no references) should serve both as a general introduction to the topic, and as a brief, non-technical summary of your main results and their implications. It should start by outlining the background to your work (why the topic is important) and the main question you have addressed (the specific problem that initiated your research), before going on to describe your new observations, main conclusions and their general implications. Because we hope that scientists across the wider microbiology community will be interested in your work, the first paragraph should be as accessible as possible, explaining essential but specialised terms

concisely. We suggest you show your summary paragraph to colleagues in other fields to uncover any problematic concepts.

Please include a data availability statement as a separate section after Methods but before references, under the heading "Data Availability". This section should inform readers about the availability of the data used to support the conclusions of your study. This information includes accession codes to public repositories (data banks for protein, DNA or RNA sequences, microarray, proteomics data etc...), references to source data published alongside the paper, unique identifiers such as URLs to data repository entries, or data set DOIs, and any other statement about data availability. At a minimum, you should include the following statement: "The data that support the findings of this study are available from the corresponding author upon request", mentioning any restrictions on availability. If DOIs are provided, we also strongly encourage including these in the Reference list (authors, title, publisher (repository name), identifier, year). For more guidance on how to write this section please see:

<http://www.nature.com/authors/policies/data/data-availability-statements-data-citations.pdf>

Please supply the figures as vector files - EPS, PDF, AI or postscript (PS) file formats (not raster or bitmap files), preferably generated with vector-graphics software (Adobe Illustrator for example). Try to ensure that all figures are non-flattened and fully editable. All images should be at least 300 dpi resolution (when figures are scaled to approximately the size that they are to be printed at) and in RGB colour format. Please do not submit Jpeg or flattened TIFF files. Please see also 'Guidelines for Electronic Submission of Figures' at the end of this letter for further detail.

Please view http://www.nature.com/authors/editorial_policies/image.html for more detailed guidelines.

We will edit your figures/tables electronically so they conform to Nature Microbiology style. If necessary, we will re-size figures to fit single or double column width. If your figures contain several parts, the parts should be labelled lower case a, b, and so on, and form a neat rectangle when assembled.

Please check the PDF of the whole paper and figures (on our manuscript tracking system) VERY CAREFULLY when you submit the revised manuscript. This will be used as the 'reference copy' to make sure no details (such as Greek letters or symbols) have gone missing during file-transfer/conversion and re-drawing.

All Supplementary Information must be submitted in accordance with the instructions in the attached Inventory of Supporting Information, and should fit into one of three categories:

1. EXTENDED DATA: Extended Data are an integral part of the paper and only data that directly contribute to the main message should be presented. These figures will be integrated into the full-text

HTML version of your paper and will be appended to the online PDF. There is a limit of 10 Extended Data figures, and each must be referred to in the main text. Each Extended Data figure should be of the same quality as the main figures, and should be supplied at a size that will allow both the figure and legend to be presented on a single legal-sized page. Each figure should be submitted as an individual .jpg, .tif or .eps file with a maximum size of 10 MB each. All Extended Data figure legends must be provided in the attached Inventory of Accessory Information, not in the figure files themselves.

2. SUPPLEMENTARY INFORMATION: Supplementary Information is material that is essential background to the study but which is not practical to include in the printed version of the paper (for example, video files, large data sets and calculations). Each item must be referred to in the main manuscript and detailed in the attached Inventory of Accessory Information. Tables containing large data sets should be in Excel format, with the table number and title included within the body of the table. All textual information and any additional Supplementary Figures (which should be presented with the legends directly below each figure) should be provided as a single, combined PDF. Please note that we cannot accept resupplies of Supplementary Information after the paper has been formally accepted unless there has been a critical scientific error.

All Extended Data must be called out in your manuscript and cited as Extended Data 1, Extended Data 2, etc. Additional Supplementary Figures (if permitted) and other items are not required to be called out in your manuscript text, but should be numerically numbered, starting at one, as Supplementary Figure 1, not SI1, etc.

3. SOURCE DATA: We strongly encourage you to provide source data for your figures whenever possible. Full-length, unprocessed gels and blots must be provided as source data for any relevant figures, and should be provided as individual PDF files for each figure containing all supporting blots and/or gels with the linked figure noted directly in the file. Numerical source data that underlie graphs are required for in vivo experiments and strongly encouraged generally. They should be provided in Excel format, one file for each relevant figure, with the linked figure noted directly in the file. They should be clearly labelled such that individual experiments and/or animals are labelled (for example, across a time course if applicable). For imaging source data, we encourage deposition to a relevant repository, such as figshare (<https://figshare.com/>) or the Image Data Resource (<https://idr.openmicroscopy.org>).

Please ensure that you retain unprocessed data and metadata files after publication, ideally archiving data in perpetuity, as these may be requested during the peer review and production process or after publication if any issues arise.

Please include any references for the Methods at the end of the reference list. Any citations in the Supplemental Information will need inclusion in a separate SI reference list.

It is a condition of publication that you include a statement before the acknowledgements naming the author to whom correspondence and requests for materials should be addressed.

Finally, we require authors to include a statement of their individual contributions to the paper -- such as experimental work, project planning, data analysis, etc. -- immediately after the acknowledgements. The statement should be short, and refer to authors by their initials. For details

please see the Authorship section of our joint Editorial policies at http://www.nature.com/authors/editorial_policies/authorship.html

We will not send your revised paper for further review if, in the editors' judgement, the referees' comments on the present version have been addressed. If the revised paper is in Nature Microbiology format, in accessible style and of appropriate length, we shall accept it for publication immediately.

Please resubmit electronically

- * the final version of the text (not including the figures) in either Word or Latex.
- * publication-quality figures. For more details, please refer to our Figure Guidelines, which is available here: <https://www.nature.com/documents/NRJs-guide-to-preparing-final-artwork.pdf>
- * Extended Data & Supplementary Information, as instructed
- * a point-by-point response to any issues raised by our referees and to any editorial suggestions.
- * any suggestions for cover illustrations, which should be provided at high resolution as electronic files. Please note that such pictures should be selected more for their aesthetic appeal than for their scientific content. I am sure you will understand that we cannot make any promise as to whether any of your suggestions might be selected for the cover of Nature Microbiology.

Please use the following link to access your home page:

{REDACTED}

* This url links to your confidential homepage and associated information about manuscripts you may have submitted or be reviewing for us. If you wish to forward this e-mail to co-authors, please delete this link to your homepage first.

Please also send the following forms as a PDF by email to microbiology@nature.com.

* Please sign and return the <http://www.nature.com/documents/snl-ltp.docx> target="_blank">Licence to Publish form .

* Or, if the corresponding author is either a Crown government employee (including Great Britain and Northern Ireland, Canada and Australia), or a US Government employee, please sign and return the <http://www.nature.com/documents/snl-ltp-crown.docx> target="_blank"> Licence to Publish form for Crown government employees, or a <http://www.nature.com/documents/snl-ltp-govus.docx> target="_blank"> Licence to Publish form for US government employees.

* Should your Article contain any items (figures, tables, images, videos or text boxes) that are the same as (or are adaptations of) items that have previously been published elsewhere and/or are owned by a third party, please note that it is your responsibility to obtain the right to use such items and to give proper attribution to the copyright holder. This includes pictures taken by professional photographers and images downloaded from the internet. If you do not hold the copyright for any

such item (in whole or part) that is included in your paper, please complete and return this [Third Party Rights Table](http://www.nature.com/documents/thirdpartyrights-origres.doc), and attach any grant of rights that you have collected.

For more information on our licence policy, please consult <http://npg.nature.com/authors>.

AUTHORSHIP

CONSORTIA -- For papers containing one or more consortia, all members of the consortium who contributed to the paper must be listed in the paper (i.e., print/online PDF). If necessary, individual authors can be listed in both the main author list and as a member of a consortium listed at the end of the paper. When submitting your revised manuscript via the online submission system, the consortium name should be entered as an author, together with the contact details of a nominated consortium representative. See <https://www.nature.com/authors/policies/authorship.html> for our authorship policy and <https://www.nature.com/documents/nr-consortia-formatting.pdf> for further consortia formatting guidelines, which should be adhered to prior to acceptance.

ORCID

Nature Microbiology is committed to improving transparency in authorship. As part of our efforts in this direction, we are now requesting that all authors identified as 'corresponding author' create and link their Open Researcher and Contributor Identifier (ORCID) with their account on the Manuscript Tracking System (MTS) prior to acceptance. ORCID helps the scientific community achieve unambiguous attribution of all scholarly contributions. For more information please visit <http://www.springernature.com/orcid>

For all corresponding authors listed on the manuscript, please follow the instructions in the link below to link your ORCID to your account on our MTS before submitting the final version of the manuscript. If you do not yet have an ORCID you will be able to create one in minutes. <https://www.springernature.com/gp/researchers/orcid/orcid-for-nature-research>

IMPORTANT: All authors identified as 'corresponding author' on the manuscript must follow these instructions. Non-corresponding authors do not have to link their ORCIDs but are encouraged to do so. Please note that it will not be possible to add/modify ORCIDs at proof. Thus, if they wish to have their ORCID added to the paper they must also follow the above procedure prior to acceptance.

To support ORCID's aims, we only allow a single ORCID identifier to be attached to one account. If you have any issues attaching an ORCID identifier to your MTS account, please contact the [Platform Support Helpdesk](http://platformsupport.nature.com/).

Nature Research journals [encourage authors to share their step-by-step experimental protocols](https://www.nature.com/nature-research/editorial-policies/reporting-standards#protocols) on a protocol sharing platform of their choice. Nature Research's Protocol Exchange is a free-to-use and open resource for protocols; protocols deposited in Protocol Exchange are citable and can be linked from the published article. More details can be found at <https://www.nature.com/protocolexchange/about>

target="new">www.nature.com/protocolexchange/about.

We hope to hear from you within two weeks; please let us know if the revision process is likely to take longer.

Reviewer Expertise:

Referee #1: Salmonella evolution, virulence, innate immunity, genomics

Referee #2: Salmonella bloodstream infections

Referee #3: Salmonella phylogenomics, genomics of pathogenesis

Reviewer Comments:

Reviewer #1 (Remarks to the Author):

I have no further comments on the manuscript.

Reviewer #2 (Remarks to the Author):

Generally, I think that the authors responded thoughtfully to the review and improved their manuscript. The authors have addressed my specific concerns by including a discussion on caveats of the "invasiveness index". I believe this work will provide a major resource to the field and recommend its publication.

Reviewer #3 (Remarks to the Author):

I am satisfied the authors have addressed all the points I made.

Final Decision Letter:

Dear Jay,

I am pleased to accept your Article "Stepwise evolution of *Salmonella* Typhimurium ST313 causing bloodstream infection in Africa." for publication in Nature Microbiology. Thank you for having chosen to submit your work to us and many congratulations.

Before your manuscript is typeset, we will edit the text to ensure it is intelligible to our wide readership and conforms to house style. We look particularly carefully at the titles of all papers to ensure that they are relatively brief and understandable.

The subeditor may send you the edited text for your approval. Once your manuscript is typeset you will receive a link to your electronic proof via email within 20 working days, with a request to make any corrections within 48 hours. If you have queries at any point during the production process then please contact the production team at rjsproduction@springernature.com. Once your paper has been scheduled for online publication, the Nature press office will be in touch to confirm the details.

Acceptance of your manuscript is conditional on all authors' agreement with our publication policies (see www.nature.com/nmicrobiolate/authors/gta/content-type/index.html). In particular your manuscript must not be published elsewhere and there must be no announcement of the work to any media outlet until the publication date (the day on which it is uploaded onto our website).

The Author's Accepted Manuscript (the accepted version of the manuscript as submitted by the author) may only be posted 6 months after the paper is published, consistent with our [self-archiving embargo](http://www.nature.com/authors/policies/license.html). Please note that the Author's Accepted Manuscript may not be released under a Creative Commons license. For Nature Research Terms of Reuse of archived manuscripts please see: <http://www.nature.com/authors/policies/license.html#terms>

To assist our authors in disseminating their research to the broader community, our SharedIt initiative provides you with a unique shareable link that will allow anyone (with or without a subscription) to read the published article. Recipients of the link with a subscription will also be able to download and

print the PDF.

As soon as your article is published (unfortunately, it is currently unclear whether the advanced online publication will be in December 2020 or early January 2021), you will receive an automated email with your shareable link.

Congratulations once again and I look forward to seeing the article published.